# Fixing the NTK: From Neural Network Linearizations to Exact Convex Programs

**Rajat Vadiraj Dwaraknath**
Stanford University
rajatvd@stanford.edu

**Tolga Ergen**
LG AI Research
tergen@lgresearch.ai

**Mert Pilanci**
Stanford University
pilanci@stanford.edu

## Abstract

Recently, theoretical analyses of deep neural networks have broadly focused on two directions: 1) Providing insight into neural network training by SGD in the limit of infinite hidden-layer width and infinitesimally small learning rate (also known as gradient flow) via the Neural Tangent Kernel (NTK), and 2) Globally optimizing the regularized training objective via cone-constrained convex reformulations of ReLU networks. The latter research direction also yielded an alternative formulation of the ReLU network, called a gated ReLU network, that is globally optimizable via efficient unconstrained convex programs. In this work, we interpret the convex program for this gated ReLU network as a Multiple Kernel Learning (MKL) model with a weighted data masking feature map and establish a connection to the NTK. Specifically, we show that for a particular choice of mask weights that do not depend on the learning targets, this kernel is equivalent to the NTK of the gated ReLU network on the training data. A consequence of this lack of dependence on the targets is that the NTK cannot perform better than the optimal MKL kernel on the training set. By using iterative reweighting, we improve the weights induced by the NTK to obtain the optimal MKL kernel which is equivalent to the solution of the exact convex reformulation of the gated ReLU network. We also provide several numerical simulations corroborating our theory. Additionally, we provide an analysis of the prediction error of the resulting optimal kernel via consistency results for the group lasso.

## 1 Introduction

Neural Networks (NNs) have become popular in various machine learning applications due to their remarkable modeling capabilities and generalization performance. However, their highly nonlinear and non-convex structure precludes an effective theoretical analysis. Therefore, developing theoretical tools to understand the fundamental mechanisms behind neural networks is still an active research topic. To tackle this problem, [1] studied the training dynamics of neural networks trained with Stochastic Gradient Descent (SGD) in a regime where each layer has infinitely many neurons and SGD uses an infinitesimally small learning rate, i.e., gradient flow. Thus, they related the training dynamics of neural networks to the training dynamics of a fixed kernel called the Neural Tangent Kernel (NTK). However, [2] showed that neurons barely move from their initial values in this regime so that neural networks fail to learn useful features from the training data. This is in contrast to their finite width counterparts, which are able to learn predictive features in practice [3]. Moreover, [4, 5] provided further theoretical and empirical evidence to show that existing kernel approaches are not able to explain the remarkable performance of finite width networks. Therefore, although NTK and similar kernel based approaches enable theoretical analysis unlike standard finite width networks, they fail to explain the effectiveness of finite width neural networks that are employed in practice.

37th Conference on Neural Information Processing Systems (NeurIPS 2023).

Recently a series of papers [6–15] introduced an analytic framework to analyze finite width neural networks by leveraging certain convex duality arguments. Particularly, they showed that the standard regularized non-convex training problem can be equivalently cast as a finite dimensional convex program. This convex approach has two major advantages over standard non-convex training: **(1)** Since the training objective is convex, one can find globally optimal parameters of the network efficiently and reliably unlike standard nonconvex training which can get stuck at a local minimum, and **(2)** As we show in this work, a class of convex reformulations can be interpreted as an instance of Multiple Kernel Learning (MKL) [16] which allows us to characterize the corresponding finite width networks by a learned data-dependent kernel that can be iteratively computed. This is in contrast to the infinite-width kernel characterization in which the NTK stays constant throughout training.

**Notation and Preliminaries.** In the paper, we use lowercase and uppercase bold letters to denote vectors and matrices respectively. We also use subscripts to denote a certain column or element. We denote the identity matrix of size $k \times k$ as $\mathbf{I}_k$. To denote the set $\{1, 2, \ldots, n\}$, we use $[n]$. We also use $\|\cdot\|_p$ and $\|\cdot\|_F$ to represent the standard $\ell_p$ and Frobenius norms. Additionally, we denote 0-1 valued indicator function and ReLU activation as $\mathbb{1}\{x \geq 0\}$ and $(x)_+ := \max\{x, 0\}$, respectively.

In this paper, we focus on analyzing the regularized training problem of ReLU networks. Particularly, we consider a two-layer ReLU network with $m$ neurons whose output function is defined as follows

$$f(\mathbf{x}, \boldsymbol{\theta}) := \sum_{j=1}^{m} \left(\mathbf{x}^T \mathbf{w}_j^{(1)}\right)_+ w_j^{(2)} = \sum_{j=1}^{m} \left(\mathbb{1}\left\{\mathbf{x}^T \mathbf{w}_j^{(1)} \geq 0\right\} \mathbf{x}^T \mathbf{w}_j^{(1)}\right) w_j^{(2)}. \tag{1}$$

where $\mathbf{w}_j^{(1)} \in \mathbb{R}^d$ and $w_j^{(2)}$ are the $j^{th}$ hidden and output layer weights, respectively and $\boldsymbol{\theta} := \{(\mathbf{w}_j^{(1)}, w_j^{(2)})\}_{j=1}^{m}$ represents all trainable parameters. Given a training data matrix $\mathbf{X} \in \mathbb{R}^{n \times d}$ and a target vector $\mathbf{y} \in \mathbb{R}^n$, we minimize the following weight decay regularized training objective

$$\min_{\mathbf{W}^{(1)}, \mathbf{w}^{(2)}} \left\|\sum_{j=1}^{m} \left(\mathbf{X}\mathbf{w}_j^{(1)}\right)_+ w_j^{(2)} - \mathbf{y}\right\|_2^2 + \lambda \sum_{j=1}^{m} \left(\left\|\mathbf{w}_j^{(1)}\right\|_2^2 + |w_j^{(2)}|^2\right), \tag{2}$$

where $\lambda > 0$ is the regularization coefficient. We also use $f(\mathbf{X}, \boldsymbol{\theta}) = \sum_{j=1}^{m} \left(\mathbf{X}\mathbf{w}_j^{(1)}\right)_+ w_j^{(2)}$ for convenience. We discuss extensions to generic loss and deeper architectures in appendix.

**Our Contributions.**

- In section 4, we show that the convex formulation of the *Gated ReLU* network is equivalent to *Multiple Kernel Learning* with a specific set of *Masking Kernels*. (Theorem 4.3).

- In section 5, we connect this formulation to the Neural Tangent Kernel by showing that the NTK is a specific weighted combination of masking kernels (Theorem 5.1).

- In Corollary 5.2, we show that on the training set, the **NTK is suboptimal when compared to the optimal kernel learned by MKL**, which is equivalent to the model learnt by our convex Gated ReLU program.

- In section 6, we also derive bounds on the prediction error of this optimal kernel and specify how to choose the regularization parameter $\lambda$ (Theorem 6.1).

## 2 Convex Optimization and the NTK

Here, we briefly review the literature on convex training and the NTK theory of neural networks.

### 2.1 Convex Programs for ReLU Networks

Even though the network in (1) has only two layers, previous studies show that (2) is a challenging optimization problem due to the non-convexity of the objective function. Thus, local search heuristics such as SGD might fail to globally optimize the training objective [17–20]. To eliminate the issues associated with the inherent non-convexity, [7] introduced an exact convex reformulation of (2) as

the following constrained optimization problem

$$\min_{\mathbf{w}_i, \mathbf{w}_i'} \left\| \sum_{i=1}^{p} \mathbf{D}_i \mathbf{X}(\mathbf{w}_i - \mathbf{w}_i') - \mathbf{y} \right\|_2^2 + \lambda \sum_{i=1}^{p} \left( \|\mathbf{w}_i\|_2 + \|\mathbf{w}_i'\|_2 \right) \text{ s.t. } \begin{array}{c} (2\mathbf{D}_i - \mathbf{I})\mathbf{X}\mathbf{w}_i \geq \mathbf{0} \\ (2\mathbf{D}_i - \mathbf{I})\mathbf{X}\mathbf{w}_i' \geq \mathbf{0} \end{array}, \forall i, \quad (3)$$

where $\mathbf{D}_i \in \mathcal{D}_{\mathbf{X}}$ are $n \times n$ binary masking diagonal matrices given by $p := |\mathcal{D}_{\mathbf{X}}|$ and

$$\mathcal{D}_{\mathbf{X}} := \left\{ \text{diag} \left( \mathbb{1} \{\mathbf{X}\mathbf{u} \geq \mathbf{0}\} \right) : \mathbf{u} \in \mathbb{R}^d \right\}. \quad (4)$$

The mask set $\mathcal{D}_{\mathbf{X}}$ can be interpreted as the set of all possible ways to separate the training data $\mathbf{X}$ by a hyperplane passing through the origin. With these masks, we can characterize a single ReLU activated neuron on the training data as follows: $(\mathbf{X}\mathbf{w}_i)_+ = \text{diag} \left( \mathbb{1} \{\mathbf{X}\mathbf{w}_i \geq \mathbf{0}\} \right) \mathbf{X}\mathbf{w}_i = \mathbf{D}_i \mathbf{X}\mathbf{w}_i$ provided that $(2\mathbf{D}_i - \mathbf{I}_n)\mathbf{X}\mathbf{w}_i \geq 0$. Therefore, by enforcing these cone constraints in (3), we maintain the masking property of the ReLU activation and parameterize the neural network as a linear function of the weights, thus make the learning problem convex. We refer the reader to [6] for more details.

Although (3) is convex and therefore eliminates the drawbacks associated with the non-convexity of (2), it might still be computationally complex to solve. Precisely, a worst-case upper-bound on the number of variables is $\mathcal{O}(n^r)$, where $r = \text{rank}(\mathbf{X}) \leq \min\{n, d\}$. Although this is still significantly better than brute-force search over $2^{mn}$ ReLU patterns, it could still be exponential in the dimension $d$. To mitigate this issue, [8] proposed a relaxation of (1) called the *gated ReLU* network

$$f_{\mathcal{G}}(\mathbf{x}, \boldsymbol{\theta}) := \sum_{j=1}^{m} \left( \mathbb{1} \left\{ \mathbf{x}^T \mathbf{g}_j \geq 0 \right\} \mathbf{x}^T \mathbf{w}_j^{(1)} \right) w_j^{(2)}, \quad (5)$$

where $\mathcal{G} := \{\mathbf{g}_j\}_{j=1}^{m}$ is a set of *gate* vectors that are also optimized throughout training. Then, the corresponding non-convex learning problem is as follows

$$\min_{\mathbf{W}^{(1)}, \mathbf{w}^{(2)}, \mathcal{G}} \left\| \sum_{j=1}^{m} \text{diag} \left( \mathbb{1} \{\mathbf{X}\mathbf{g}_j \geq \mathbf{0}\} \right) \mathbf{X}\mathbf{w}_j^{(1)} w_j^{(2)} - \mathbf{y} \right\|_2^2 + \lambda \sum_{j=1}^{m} \left( \left\| \mathbf{w}_j^{(1)} \right\|_2^2 + |w_j^{(2)}|^2 \right). \quad (6)$$

By performing this relaxation, we decouple the dependence between the indicator function and the linear term in the exact ReLU network (1). To express the equivalent convex optimization problem corresponding to the gated ReLU network, we introduce the notion of complete gate sets.

**Definition 2.1.** *A gate set $\mathcal{G}$ is complete with respect to a dataset $\mathbf{X}$ if the corresponding set of hyperplane arrangements covers all possible arrangement patterns for $\mathbf{X}$ defined in (4), i.e.,*

$$\{\text{diag} \left( \mathbb{1} \{\mathbf{X}\mathbf{g}_j \geq \mathbf{0}\} \right) : \mathbf{g}_j \in \mathcal{G}\} = \mathcal{D}_{\mathbf{X}}.$$

*Additionally, $\mathcal{G}$ is minimally complete if $|\mathcal{G}| = |\mathcal{D}_{\mathbf{X}}| = p$.*

Now, with this relaxation, [8] showed that the optimal value for (6) can always be achieved by choosing $\mathcal{G}$ to be a complete gate set. Therefore, by working only with complete gate sets, we can modify (6) to only optimize over the network parameters $\mathbf{W}^{(1)}$ and $\mathbf{w}^{(2)}$

$$\min_{\mathbf{W}^{(1)}, \mathbf{w}^{(2)}} \left\| \sum_{j=1}^{m} \text{diag} \left( \mathbb{1} \{\mathbf{X}\mathbf{g}_j \geq \mathbf{0}\} \right) \mathbf{X}\mathbf{w}_j^{(1)} w_j^{(2)} - \mathbf{y} \right\|_2^2 + \lambda \sum_{j=1}^{m} \left( \left\| \mathbf{w}_j^{(1)} \right\|_2^2 + |w_j^{(2)}|^2 \right). \quad (7)$$

Additionally, [21] also showed that we can set $m = p$ without loss of generality. Then, [8] showed that the equivalent convex optimization problem for (6) and also (7) in the complete gate set setting is

$$\min_{\mathbf{w}_i} \left\| \sum_{i=1}^{p} \mathbf{D}_i \mathbf{X}\mathbf{w}_i - \mathbf{y} \right\|_2^2 + \lambda \sum_{i=1}^{p} \|\mathbf{w}_i\|_2. \quad (8)$$

Notice that (8) is a least squares problem with *group lasso* regularization [22]. Therefore, this relaxation for (3) can be efficiently optimized via convex optimization solvers. Furthermore, [8] proved that after solving the relaxed problem (8), one can construct an equivalent ReLU network from a gated ReLU network via a convex optimization based procedure called *cone decomposition*. We discuss the computational complexity of this approach in Section E of the supplementary material.

## 2.2 The Neural Tangent Kernel

Previous works [1, 2, 5, 23, 24] characterized the training dynamics of SGD with infinitesimally small learning rates on neural networks in the *infinite-width* limit, i.e., as $m \to \infty$, via the NTK. [25, 26] also present analyses in this regime via a mean-field approach. In this section, we provide a brief overview of this theory and refer the reader to [1, 27, 28] for more details. The main idea behind the NTK theory is to approximate the neural network model function $f(\mathbf{x}, \boldsymbol{\theta})$ by *linearizing it* with respect to the parameters $\boldsymbol{\theta}$ around its initialization $\boldsymbol{\theta}_0$:

$$\hat{f}(\mathbf{x}, \boldsymbol{\theta}) \approx f(\mathbf{x}, \boldsymbol{\theta}_0) + \nabla_{\boldsymbol{\theta}} f(\mathbf{x}, \boldsymbol{\theta}_0)^T (\boldsymbol{\theta} - \boldsymbol{\theta}_0).$$

The authors of [1] show that if $f$ is a neural network (with appropriately scaled output), and parameters $\boldsymbol{\theta}$ initialized as i.i.d standard Gaussians, the linearization $\hat{f}$ better approximates $f$ in the infinite-width limit. We can interpret the linearized model $\hat{f}$ as a kernel method with a feature map given by $\boldsymbol{\phi}(\mathbf{x}) = \nabla_{\boldsymbol{\theta}} f(\mathbf{x}, \boldsymbol{\theta}_0)$. The corresponding kernel induced by this feature map is termed as the NTK. Note that this is a random kernel since it depends on the random initialization of the parameters denoted as $\boldsymbol{\theta}_0$. The main result of [1] in the simplified case of two-layer neural networks is that, in the infinite-width limit this kernel approaches a fixed deterministic limit given by $H(\mathbf{x}, \mathbf{x}') := \mathbb{E}[\nabla_{\hat{\boldsymbol{\theta}}} f(\mathbf{x}, \hat{\boldsymbol{\theta}}_0)^T \nabla_{\hat{\boldsymbol{\theta}}} f(\mathbf{x}', \hat{\boldsymbol{\theta}}_0)]$, where $\hat{\boldsymbol{\theta}}$ corresponds to the parameters of a single neuron. Furthermore, [1] show that in this infinite limit, SGD with an infinitesimally small learning rate is equivalent to performing kernel regression with the fixed NTK.

To link the convex formulation (8) with NTK theory, we first present a scaled version of the gated ReLU network in (5) as follows

$$\tilde{f}_{\mathcal{G}}(\mathbf{x}, \boldsymbol{\theta}) := \frac{1}{\sqrt{2m}} \sum_{j=1}^{m} \left( \mathbb{1}\left\{ \mathbf{x}^T \mathbf{g}_j \geq \mathbf{0} \right\} \mathbf{x}^T \mathbf{w}_j^{(1)} \right) w_j^{(2)}. \tag{9}$$

In the next lemma, we provide the infinite width NTK of the scaled gated ReLU network in (9).

**Lemma 2.2.** [1] *The infinite width NTK of the gated ReLU network* (9) *with i.i.d gates sampled as* $\mathbf{g}_j \sim \mathcal{N}(\mathbf{0}, \mathbf{I_d})$ *and randomly initialized parameters as* $\mathbf{w}_j^{(1)} \sim \mathcal{N}(\mathbf{0}, \mathbf{I_d})$ *and* $w_j^{(2)} \sim \mathcal{N}(0, 1)$ *is*

$$H(\mathbf{x}, \mathbf{x}') := \frac{1}{2\pi} \left( \pi - \arccos\left( \frac{\mathbf{x}^T \mathbf{x}'}{\|\mathbf{x}\|_2 \|\mathbf{x}'\|_2} \right) \right) \mathbf{x}^T \mathbf{x}'. \tag{10}$$

Additionally, we introduce a reparameterization of the standard ReLU network (1) with $\boldsymbol{\theta} := \left\{ \left( \mathbf{w}_j^{(+)}, \mathbf{w}_j^{(-)} \right) \right\}_{j=1}^{m}$, with $\mathbf{w}_j^{(+)}, \mathbf{w}_j^{(-)} \in \mathbb{R}^d$ which can still represent all the functions that (1) can

$$f_r(\mathbf{x}, \boldsymbol{\theta}) := \frac{1}{\sqrt{2m}} \sum_{j=1}^{m} \left( \mathbf{x}^T \mathbf{w}_j^{(+)} \right)_+ - \left( \mathbf{x}^T \mathbf{w}_j^{(-)} \right)_+. \tag{11}$$

**Lemma 2.3.** *The gated ReLU network* (9) *and the reparameterized ReLU network* (11) *have the same infinite width NTK given by Lemma* 2.2.

Next, we present an equivalence between the gated ReLU network and the MKL model [16, 29].

## 3 Multiple Kernel Learning and Group Lasso

The Multiple Kernel Learning (MKL) model [16, 29] is an extension of the standard kernel method that learns an optimal data-dependent kernel as a convex combination of a set of fixed kernels and then performs regression with this learned kernel. We provide a brief overview of the MKL setting based on the exposition in [30]. Consider a set of $p$ kernels given by corresponding feature maps $\boldsymbol{\phi}_i : \mathbb{R}^d \to \mathbb{R}^{d_i}$. Given $n$ training samples $\mathbf{X} \in \mathbb{R}^{n \times d}$ with targets $\mathbf{y} \in \mathbb{R}^n$, we define the feature matrices on this data by stacking the feature vectors as $\boldsymbol{\Phi}_i := [\boldsymbol{\phi}_i(\mathbf{x}_1)^T; \dots; \boldsymbol{\phi}_i(\mathbf{x}_n)^T] \in \mathbb{R}^{n \times d_i}$. Then, the corresponding $n \times n$ kernel matrices are given by $\mathbf{K}_i := \boldsymbol{\Phi}_i \boldsymbol{\Phi}_i^T$. A convex combination of these kernels can be written as $\mathbf{K}(\boldsymbol{\eta}) := \sum_{i=1}^{p} \eta_i \mathbf{K}_i$ where $\boldsymbol{\eta} \in \Delta_p := \{\boldsymbol{\eta} : \mathbf{1}^T \boldsymbol{\eta} = 1, \boldsymbol{\eta} \geq \mathbf{0}\}$ is a set of weights in the unit simplex. By noticing that the feature map corresponding to $\mathbf{K}(\boldsymbol{\eta})$

---

[1]All the proofs and derivations are presented in the supplementary material.

is obtained by taking a weighted concatenation of $\phi_i$ with weights $\sqrt{\eta_i}$, we can write the MKL optimization problem in terms of the feature matrices as

$$\min_{\boldsymbol{\eta}\in\Delta_p, \mathbf{v}_i\in\mathbb{R}^{d_i}} \left\| \sum_{i=1}^{p} \sqrt{\eta_i}\boldsymbol{\Phi}_i\mathbf{v}_i - \mathbf{y} \right\|_2^2 + \hat{\lambda}\sum_{i=1}^{p} \|\mathbf{v}_i\|_2^2, \tag{12}$$

where $\hat{\lambda} > 0$ is a regularization coefficient. For a set of fixed weights $\boldsymbol{\eta}$, the optimal objective value of the kernel regression problem over $\mathbf{v}$ is proportional to $\mathbf{y}^T(\sum_{i=1}^{p}\eta_i\mathbf{K}_i + \hat{\lambda}\mathbf{I_n})^{-1}\mathbf{y}$ up to constant factors [16]. Thus, the MKL problem can be equivalently written as the following problem

$$\min_{\boldsymbol{\eta}\in\Delta_p} \quad \mathbf{y}^T\left(\mathbf{K}(\boldsymbol{\eta}) + \hat{\lambda}\mathbf{I_n}\right)^{-1}\mathbf{y}. \tag{13}$$

In this formulation, we can interpret MKL as finding the optimal data-dependent kernel that can be expressed as a convex combination of the fixed kernels given by $\mathbf{K}_i$. In the next section, we link this kernel learning formulation with the convex group lasso problem in (8).

## 3.1 Equivalence to Group Lasso

We first show that the MKL problem in (12) can be equivalently stated as a group lasso problem.

**Lemma 3.1** ([16, 29])**.** *The MKL problem* (12) *is equivalent to the following kernel regression problem using a uniform combination of the fixed kernels with squared group lasso regularization where the groups are given by parameters corresponding to each feature map*

$$\min_{\mathbf{w}_i\in\mathbb{R}^{d_i}} \quad \left\| \sum_{i=1}^{p} \boldsymbol{\Phi}_i\mathbf{w}_i - \mathbf{y} \right\|_2^2 + \hat{\lambda}\left(\sum_{i=1}^{p} \|\mathbf{w}_i\|_2\right)^2. \tag{14}$$

We now present a short derivation of this equivalence. Using the variational formulation of the squared group $\ell_1$-norm [31]

$$\left(\sum_{i=1}^{p} \|\mathbf{w}_i\|_2\right)^2 = \min_{\boldsymbol{\eta}\in\Delta_p} \sum_{i=1}^{p} \frac{\|\mathbf{w}_i\|_2^2}{\eta_i},$$

we can rewrite the group lasso problem (14) as a joint minimization problem over both the parameters $\mathbf{w}$ and regularization weights $\boldsymbol{\eta}$ as follows

$$\min_{\boldsymbol{\eta}\in\Delta_p} \quad \min_{\mathbf{w}_i\in\mathbb{R}^{d_i}} \quad \left\| \sum_{i=1}^{p} \boldsymbol{\Phi}_i\mathbf{w}_i - \mathbf{y} \right\|_2^2 + \hat{\lambda}\sum_{i=1}^{p} \frac{\|\mathbf{w}_i\|_2^2}{\eta_i}.$$

Finally, with a change of variables given by $\mathbf{v}_i = \mathbf{w}_i/\sqrt{\eta_i}$, we recover the MKL problem (12). We note that the MKL problem (12) is also equivalent to the following standard group lasso problem

$$\min_{\mathbf{w}_i\in\mathbb{R}^{d_i}} \quad \left\| \sum_{i=1}^{p} \boldsymbol{\Phi}_i\mathbf{w}_i - \mathbf{y} \right\|_2^2 + \lambda\sum_{i=1}^{p} \|\mathbf{w}_i\|_2. \tag{15}$$

This is due to the fact that squared and standard group lasso problems have the same regularization paths [31], so (14) and (15) are equivalent when $\hat{\lambda} = \frac{\lambda}{\sum_{i=1}^{p}\|\mathbf{w}_i^*\|_2}$, where $\mathbf{w}^*$ is the solution to (14).

## 3.2 Solving Group Lasso by Iterative Reweighting

Previously, we used a variational formulation of the squared group $\ell_1$-norm to show equivalences to MKL. Now, we present the Iteratively Reweighted Least Squares (IRLS) algorithm [32–35] to solve the group lasso problem (15) using the following variational formulation of the group $\ell_1$-norm [34]

$$\sum_{i=1}^{p} \|\mathbf{w}_i\|_2 = \min_{\boldsymbol{\eta}\in\mathbb{R}_+^p} \frac{1}{2}\sum_{i=1}^{p} \left(\frac{\|\mathbf{w}_i\|_2^2}{\eta_i} + \eta_i\right).$$

Based on this, we rewrite the group lasso problem (15) as the following minimization problem

$$\min_{\boldsymbol{\eta} \in \mathbb{R}_+^p} \min_{\mathbf{w}_i \in \mathbb{R}^{d_i}} \left\| \sum_{i=1}^p \boldsymbol{\Phi}_i \mathbf{w}_i - \mathbf{y} \right\|_2^2 + \frac{\lambda}{2} \sum_{i=1}^p \left( \frac{\|\mathbf{w}_i\|_2^2}{\eta_i} + \eta_i \right).$$

Since the objective is jointly convex in $(\boldsymbol{\eta}, \mathbf{w})$, it can be solved using alternating minimization [33]. Particularly, note that the inner minimization problem in $\mathbf{w}_i$'s is simply a $\ell_2$ regularized least squares problem with different regularization strengths for each group and this can be solved in closed form

$$\min_{\mathbf{w}_i \in \mathbb{R}^{d_i}} \left\| \sum_{i=1}^p \boldsymbol{\Phi}_i \mathbf{w}_i - \mathbf{y} \right\|_2^2 + \lambda \sum_{i=1}^p \frac{\|\mathbf{w}_i\|_2^2}{\eta_i}. \tag{16}$$

The outer problem in $\boldsymbol{\eta}$ is also directly solved by setting $\eta_i = \|\mathbf{w}_i\|_2$ [34]. To avoid convergence issues and instability around $\eta_i = 0$, we approximate the reweighting by adding a small positive constant $\epsilon$. We use this procedure to solve the group lasso formulation of the gated ReLU network (8) by setting $\boldsymbol{\Phi}_i = \mathbf{D}_i \mathbf{X}$. A detailed description is provided Algorithm 1. For further details regarding convergence, we refer the reader to [32–34].

---

**Algorithm 1** Iteratively Reweighted Least Squares (IRLS) for gated ReLU and ReLU networks

---

1: Set iteration count $k \leftarrow 0$
2: Initialize weights $\eta_i^{(0)}$
3: Set $\boldsymbol{\Phi}_i := \mathbf{D}_i \mathbf{X}, \forall \, \mathbf{D}_i \in \mathcal{D}_\mathbf{X}$
4: **while** not converged and $k \leq$ max iteration count **do**
5:     Solve the weighted $\ell_2$ regularized least squares problem:

$$\left\{ \mathbf{w}_i^{(k)} \right\}_i = \operatorname*{argmin}_{\{\mathbf{w}_i\}_i} \left\| \sum_{i=1}^p \boldsymbol{\Phi}_i \mathbf{w}_i - \mathbf{y} \right\|_2^2 + \lambda \sum_{i=1}^p \frac{\|\mathbf{w}_i\|_2^2}{\eta_i^{(k)}}$$

6:     Update the weights: $\eta_i^{(k+1)} = \sqrt{\left\| \mathbf{w}_i^{(k)} \right\|_2 + \epsilon}$
7:     Increment iteration count: $k \leftarrow k + 1$
8: **end while**
9: **Optional:** Convert the gated ReLU network to a ReLU network (see Section E for details)

---

## 4 Gated ReLU as MKL with Masking Kernels

Motivated by the MKL interpretation of group lasso, we return to the convex reformulation (8) of the gated ReLU network. Notice that this problem has the same structure as the MKL equivalent group lasso problem (15) with a specific set of feature maps that we define below.

**Definition 4.1.** *The masking feature maps $\boldsymbol{\phi}_j : \mathbb{R}^d \to \mathbb{R}^d$ generated by a fixed set of gates $\mathcal{G}$ are defined as $\boldsymbol{\phi}_j(\mathbf{x}) = \mathbb{1}\left\{\mathbf{x}^T \mathbf{g}_j \geq 0\right\} \mathbf{x}$.*

These feature maps can be interpreted as simply passing the input unchanged if it lies in the positive halfspace of the corresponding gate vector $\mathbf{g}_j$, i.e., $\mathbf{x}^T \mathbf{g}_j \geq 0$, and returning zero if the input does not lie in this halfspace. Since $\operatorname{diag}\left(\mathbb{1}\left\{\mathbf{X}\mathbf{g}_j \geq \mathbf{0}\right\}\right) \in \mathcal{D}_\mathbf{X}, \forall \, \mathbf{g}_j \in \mathcal{G}$ holds for an arbitrary gate set $\mathcal{G}$, we can conveniently express the corresponding feature matrices of these masking feature maps on the data $\mathbf{X}$ in terms of fixed diagonal data masks as $\boldsymbol{\Phi}_j = \mathbf{D}_j \mathbf{X}$, where $\mathbf{D}_j = \operatorname{diag}\left(\mathbb{1}\left\{\mathbf{X}\mathbf{g}_j \geq \mathbf{0}\right\}\right)$. Similarly, the corresponding masking kernel matrices take the form $\mathbf{K}_j = \mathbf{D}_j \mathbf{X} \mathbf{X}^T \mathbf{D}_j$. Note that for an arbitrary set of gates, the generated masking feature matrices on $\mathbf{X}$ may not cover the entire set of possible masks $\mathcal{D}_\mathbf{X}$. Additionally, multiple gate vectors can result in identical masks if $\operatorname{diag}\left(\mathbb{1}\left\{\mathbf{X}\mathbf{g}_i \geq \mathbf{0}\right\}\right) = \operatorname{diag}\left(\mathbb{1}\left\{\mathbf{X}\mathbf{g}_j \geq \mathbf{0}\right\}\right)$ for $i \neq j$ leading to degenerate feature matrices. However, if we work with minimally complete gate sets as defined in Definition 2.1, we can rectify these issues.

**Lemma 4.2.** *For a minimally complete gate set $\mathcal{G}$ defined in Definition 2.1, we can uniquely associate a gate vector $\mathbf{g}_i$ to each data mask $\mathbf{D}_i \in \mathcal{D}_\mathbf{X}$ such that $\mathbf{D}_i = \operatorname{diag}\left(\mathbb{1}\left\{\mathbf{X}\mathbf{g}_i \geq \mathbf{0}\right\}\right), \forall i \in [p]$.*

Consequently, for minimally complete gate sets $\mathcal{G}$, the generated masking feature matrices $\forall i \in [p]$ can be expressed as $\mathbf{\Phi}_i = \mathbf{D}_i \mathbf{X}$ and the masking kernel matrices take the form $\mathbf{K}_i = \mathbf{D}_i \mathbf{X} \mathbf{X}^T \mathbf{D}_i$. In the context of the gated ReLU problem (7), since $\mathcal{G}$ is complete, we can replace it with a minimally complete subset of $\mathcal{G}$ without loss of generality since [21] showed that increasing $m$ beyond $p$ cannot reduce the value of the regularized training objective in (7). We are now ready to combine the MKL-group lasso equivalence with the convex reformulation of the gated ReLU network to present the following characterization of the nonconvex gated ReLU learning problem.

**Theorem 4.3.** *The non-convex gated ReLU problem* (7) *with a minimally complete gate set $\mathcal{G}$ is equivalent to performing multiple kernel learning* (12) *with the masking feature maps generated by $\mathcal{G}$*

$$\min_{\boldsymbol{\eta} \in \Delta_p, \mathbf{v}_i \in \mathbb{R}^d} \quad \left\| \sum_{i=1}^{p} \sqrt{\eta_i} \mathbf{D}_i \mathbf{X} \mathbf{v}_i - \mathbf{y} \right\|_2^2 + \hat{\lambda} \sum_{i=1}^{p} \|\mathbf{v}_i\|_2^2 .$$

This theorem implies that the gated ReLU network finds the optimal combination of linear models restricted to the different masked datasets $\mathbf{D}_i \mathbf{X}$ generated by the gates. By optimizing with all possible data maskings, we obtain the best possible gated ReLU network. From the kernel perspective, we have characterized the problem of finding an optimal finite width gated ReLU network as learning a data-dependent kernel and then performing kernel regression. This is in contrast to the NTK theory where the training of an infinite width network by gradient flow is characterized by regression with a constant kernel that is not learned from data. We further explore this connection below.

# 5 NTK as a Weighted Masking Kernel

We now connect the NTK of a gated ReLU network with the masking kernels generated by its gates.

**Theorem 5.1.** *Let $\mathbf{K}_{\mathcal{G}} (\tilde{\boldsymbol{\eta}}) \in \mathbb{R}^{n \times n}$ be the weighted masking kernel obtained by taking a convex combination of the masking feature maps generated by a minimally complete gate set $\mathcal{G}$ with weights given by $\tilde{\eta}_i = \mathbb{P}[\mathrm{diag}\,(\mathbb{1}\,\{\mathbf{X}\mathbf{h} \geq \mathbf{0}\}) = \mathbf{D}_i]$ where $\mathbf{h} \sim \mathcal{N}(\mathbf{0}, \mathbf{I_d})$ and let $\mathbf{H} \in \mathbb{R}^{n \times n}$ be the infinite width NTK of the gated ReLU network* (10) *evaluated on the training data, i.e., the $ij^{th}$ entry of $\mathbf{H}$ is defined as $\mathbf{H}_{ij} := H(\mathbf{x}_i, \mathbf{x}_j)$. Then, $\mathbf{K}_{\mathcal{G}} (\tilde{\boldsymbol{\eta}}) = \mathbf{H}$.*

A rough sketch of the proof of this theorem is to express the matrix $\mathbf{H}$ as an expectation of indicator random variables using the definition of the NTK [1]. Then, by conditioning on the event that these indicators equal the masks $\mathbf{D}_i$, we can express the NTK as a convex combination of the masking kernels $\mathbf{K}_i$. The weights end up being precisely the probabilities that are described in Theorem 5.1. A detailed proof is provided in the supplementary material.

This theorem implies that the outputs of the gated ReLU network obtained via (16) with regularization weights $\tilde{\boldsymbol{\eta}}$ on the training data is identical to that of kernel ridge regression with the NTK.

**Corollary 5.2.** *Let $\tilde{\mathbf{w}}$ be the solution to* (16) *with feature matrices $\mathbf{\Phi}_i = \mathbf{D}_i \mathbf{X}$ and regularization weights $\tilde{\eta}_i = \mathbb{P}[\mathrm{diag}\,(\mathbb{1}\,\{\mathbf{X}\mathbf{h} \geq \mathbf{0}\}) = \mathbf{D}_i]$ where $\mathbf{h} \sim \mathcal{N}(\mathbf{0}, \mathbf{I_d})$ and let $\tilde{\mathbf{y}} = \mathbf{H}(\mathbf{H} + \lambda \mathbf{I_n})^{-1} \mathbf{y}$ be the outputs of kernel ridge regression with the NTK on the training data. Then, $\sum_{i=1}^{p} \mathbf{D}_i \mathbf{X} \tilde{\mathbf{w}}_i = \tilde{\mathbf{y}}$.*

Since $\mathcal{G}$ is minimally complete, the weights in Theorem 5.1 satisfy $\sum_{i=1}^{p} \tilde{\eta}_i = 1$. In other words, $\tilde{\boldsymbol{\eta}} \in \Delta_p$. Therefore, we can interpret Theorem 5.1 as follows – the NTK evaluated on the training data lies in the convex hull of all possible masking kernels of the training data. So $\mathbf{H}$ lies in the feasible set of kernels for MKL using these masking kernels. By Theorem 4.3, we can find the optimal kernel in this set by solving the group lasso problem (8) of the gated ReLU network. **Therefore, we can interpret solving** (8) **as fixing the NTK by learning an improved data-dependent kernel.**

**Remark 5.3** (**Suboptimality of NTK**). *Note that the weights $\tilde{\boldsymbol{\eta}}$ do depend on the training data $\mathbf{X}$, but do not depend on the target labels $\mathbf{y}$. Since MKL learns the optimal kernel using both $\mathbf{X}$ and $\mathbf{y}$, the NTK still cannot perform better than the optimal MKL kernel on the training set. Thus, we fix the NTK.*

# 6 Analysis of Prediction Error

In this section, we present an analysis of the in-sample prediction error for the gated ReLU network given in (5) along the lines of existing consistency results [36, 37] for the group lasso problem (8).

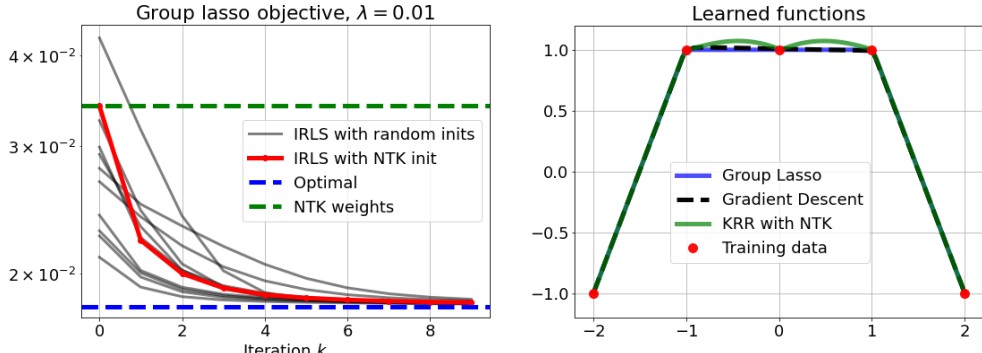

Figure 1: Plot of objective value of problem (8) which is solved using IRLS (algorithm 1) for a toy 1D dataset with $n = 5$. The iterates are compared to the optimal value obtained by solving (8) using CVXPY (blue). Notice that the solution to (16) with regularization weights given by the NTK weights $\tilde{\boldsymbol{\eta}}$ from Theorem 5.1 (green) is sub-optimal for problem (8), and running IRLS by initializing with these weights (red) converges to the optimal objective value. We also include plots of IRLS initialized with random weights (black). The right plot shows the corresponding learned functions. The blue curve shows the output of the solution to the group lasso problem (8) after performing a cone decomposition to obtain a ReLU network. The dashed curve shows the output of running gradient descent (GD) on a ReLU network with 100 neurons. The green curve is the result of kernel ridge regression (KRR) with the NTK. **We observe that our reweighted kernel method (Group Lasso) produces an output that matches the output of the NN trained via GD. In contrast, NTK produces an erroneous smooth function due to the infinite width approximation.**

We assume that the data is generated by a noisy ReLU neural network model $\mathbf{y} = f(\mathbf{X}, \boldsymbol{\theta}^*) + \epsilon$ where $f$ is the ReLU network defined in (1) with true parameters $\boldsymbol{\theta}^*$ and the noise is distributed as $\epsilon \sim \mathcal{N}\left(0, \sigma^2 \mathbf{I}_n\right)$. By the seminal universal approximation theorem of [38], this model is able to capture a broad class of ground-truth functions by using a ReLU network with enough neurons. We can transform $\boldsymbol{\theta}^*$ to the weights $\mathbf{w}^*$ in the convex ReLU model and write $f(\mathbf{X}, \boldsymbol{\theta}^*) = \sum_{i=1}^{p} \mathbf{D}_i \mathbf{X} \mathbf{w}_i^*$. We denote by $\hat{\mathbf{w}}$ the solution of the group lasso problem (8) for the gated ReLU network with an additional $\frac{1}{n}$ factor on the loss to simplify derivations,

$$\hat{\mathbf{w}} = \underset{\mathbf{w}_i \in \mathbb{R}^d}{\operatorname{argmin}} \quad \frac{1}{n} \left\| \sum_{i=1}^{p} \mathbf{D}_i \mathbf{X} \mathbf{w}_i - \mathbf{y} \right\|_2^2 + \lambda \sum_{i=1}^{p} \|\mathbf{w}_i\|_2. \tag{17}$$

We now present our main theorem which bounds the prediction error of the gated ReLU network obtained from the solution $\hat{\mathbf{w}}$ below.

**Theorem 6.1** (Prediction Error of Gated ReLU). *For some $t > 0$, let the regularization parameter in (17) be $\lambda = t\sigma\|\mathbf{X}\|_F/n$. Then, with probability at least $1 - 2e^{-t^2/8}$, we have*

$$\frac{1}{n} \left\| f_{\mathcal{G}}\left(\mathbf{X}, \hat{\boldsymbol{\theta}}\right) - f\left(\mathbf{X}, \boldsymbol{\theta}^*\right) \right\|_2^2 \le 2\lambda \sum_{i=1}^{p} \|\mathbf{w}_i^*\|_2$$

*where $f_{\mathcal{G}}\left(\mathbf{X}, \hat{\boldsymbol{\theta}}\right) = \sum_{i=1}^{p} \mathbf{D}_i \mathbf{X} \hat{\mathbf{w}}_i$ are the predictions of the gated ReLU network obtained from $\hat{\mathbf{w}}$.*

The proof closely follows the analysis of the regular lasso problem presented in [37], but we extend it to the specific case of the group lasso problem corresponding to the gated ReLU network and leverage the masking structure of the lifted data matrix to obtain simplified bounds.

## 7 Experiments

Here, we empirically corroborate our theoretical results via experiments on several datasets.[2]

---

[2]We provide additional details in Section B.

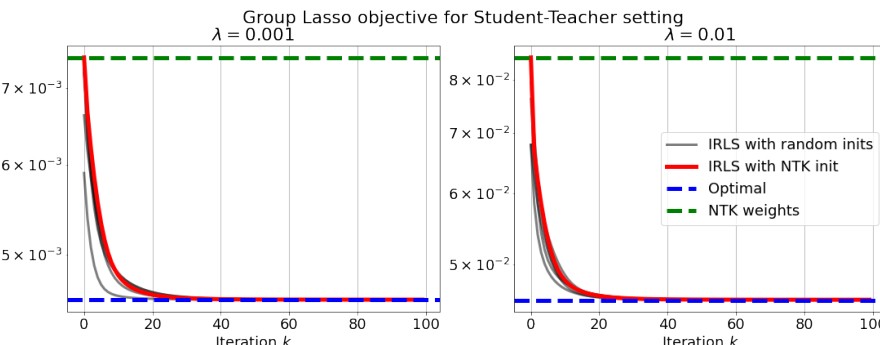

Figure 2: Plot of the **group lasso objective** in the student-teacher setting with $d = 5$. Training data is generated using a teacher network with width $m = 10$. The NTK weights $\tilde{\boldsymbol{\eta}}$ are estimated using Monte Carlo sampling, and are again sub-optimal. IRLS initialized with these weights successfully fixes the NTK and converges to the optimal weights.

**1D datasets.** For the 1D experiments in Figure 1, we add a second data dimension with value equal to 1 for all data points to simulate a bias term in the first layer of the gated ReLU network. Also, in this case we can enumerate all $2n$ data masks $\mathbf{D}_i$ directly. We use the cone decomposition procedure described in [8] to obtain a ReLU network from the gated ReLU network obtained by solving (8). We also train a 100 neuron ReLU network using gradient descent (GD) and compare the learned output functions in the right plot in Figure 1. Exact details can be found in the supplementary material.

**Student-teacher setting.** We generate the training data by sampling $\mathbf{X} \sim \mathcal{N}(\mathbf{0}, \mathbf{I_n})$ and computing the targets $\mathbf{y}$ using a fixed, randomly initialized gated ReLU teacher network. In Figure 2, $n = 10$, $d = 5$, and we use a teacher network with width $m = 10$, with gates and parameters randomly drawn from independent standard multivariate Gaussians. To solve the convex formulation (8), we estimate $\mathcal{D}_{\mathbf{X}}$ by randomly sampling unique hyperplane arrangements.

**Computing the NTK weights $\tilde{\eta}$.** The weights induced by the NTK are

Table 1: Test accuracies for UCI experiments with $75\% - 25\%$ training-test split. Our approach achieves either higher or the same accuracy for 26 out of 33 datasets.

| Dataset | $n$ | $d$ | NTK | Ours (Alg 1) |
|---|---|---|---|---|
| acute-inflammation | 120 | 6 | 1.000 | 1.000 |
| acute-nephritis | 120 | 6 | 1.000 | 1.000 |
| balloons | 16 | 4 | 0.75 | 0.75 |
| blood | 748 | 4 | 0.524 | **0.583** |
| breast-cancer | 286 | 9 | 0.417 | **0.625** |
| breast-cancer-wisc-prog | 699 | 9 | 0.96 | **0.966** |
| breast-cancer-wisc-diag | 569 | 30 | **0.965** | 0.915 |
| breast-cancer-wisc-prog | 198 | 33 | **0.7** | 0.66 |
| congressional-voting | 435 | 16 | 0.266 | 0.266 |
| conn-bench-sonar-mines-rocks | 208 | 60 | 0.635 | **0.712** |
| credit-approval | 690 | 15 | 0.838 | **0.844** |
| cylinder-bands | 512 | 35 | 0.773 | **0.82** |
| echocardiogram | 131 | 10 | 0.758 | **0.788** |
| fertility | 100 | 9 | 0.76 | 0.76 |
| haberman-survival | 306 | 3 | 0.481 | **0.532** |
| heart-hungarian | 294 | 12 | 0.743 | **0.878** |
| hepatitis | 155 | 19 | **0.923** | 0.897 |
| ilpd-indian-liver | 583 | 9 | 0.432 | **0.555** |
| ionosphere | 351 | 33 | 0.955 | **0.966** |
| mammographic | 961 | 5 | 0.783 | **0.792** |
| molec-biol-promoter | 106 | 57 | 0.63 | **0.815** |
| musk-1 | 476 | 166 | 0.782 | **0.866** |
| oocytes_trisopterus_nucleus_2f | 912 | 25 | **0.781** | 0.759 |
| parkinsons | 195 | 22 | 0.939 | **0.959** |
| pima | 768 | 8 | 0.552 | **0.599** |
| pittsburg-bridges-T-OR-D | 102 | 7 | 0.731 | **0.846** |
| planning | 182 | 12 | 0.435 | **0.543** |
| statlog-australian-credit | 690 | 14 | **1.0** | 0.74 |
| statlog-german-credit | 1000 | 24 | 0.512 | **0.576** |
| statlog-heart | 270 | 13 | **0.779** | 0.765 |
| tic-tac-toe | 958 | 9 | 1.0 | 1.0 |
| trains | 10 | 29 | 0.667 | 0.667 |
| vertebral-column-2clases | 310 | 6 | **0.821** | 0.731 |
| Higher (or same) accuracy | | | 14/33 | 26/33 |

given as in Theorem 5.1 by $\tilde{\eta}_i = \mathbb{P}[\text{diag}\,(\mathbb{1}\,\{\mathbf{Xh} \geq \mathbf{0}\}) = \mathbf{D}_i]$ where $\mathbf{h} \sim \mathcal{N}(\mathbf{0}, \mathbf{I_d})$ is a standard multivariate Gaussian vector. These probabilities can be interpreted either as the orthant probabilities of the multivariate Gaussians given by $(2\mathbf{D}_i - \mathbf{I_n})\,\mathbf{Xh}$ or as the solid angle of the cones given by $\{\mathbf{u} \in \mathbb{R}^d : (2\mathbf{D}_i - \mathbf{I_n})\,\mathbf{Xu} \geq \mathbf{0}\}$. Closed form expressions exist for $d = 2, 3$, and [39, 40] present approximating schemes for higher dimensions. We calculate these weights exactly for the 1D example presented in Figure 1, and estimate them using Monte Carlo sampling for the student-teacher example in Figure 2.

**Fixing the NTK weights by IRLS.** We solve (16) with $\mathbf{\Phi}_i = \mathbf{D}_i\mathbf{X}$ and regularization weights given by the NTK weights $\tilde{\boldsymbol{\eta}}$ to obtain the solution $\tilde{\mathbf{w}}$. We use efficient least squares solvers from [41, 42]. By Theorem 5.1, this corresponds to choosing the weights $\boldsymbol{\eta}$ in the MKL problem (13) such that the resulting kernel matrix is equal to the NTK matrix $\mathbf{H}$. We find the exact solution of the group lasso problem (8) using CVXPY. Comparing the optimal value of the group lasso problem (8) with the objective value of $\tilde{\mathbf{w}}$ (given by the green line) in Figures 1 and 2, we observe that the NTK weighted solution is sub-optimal. This means that $\mathbf{H}$ is not the optimal kernel that would be learnt by MKL (which is expected since $\mathbf{H}$ has no dependence on the targets $\mathbf{y}$). By applying IRLS initialized with the NTK weights, we *fix* the NTK and find the weights of the optimal MKL kernel. In Figures 1 and 2, we observe that IRLS converges to the solution of the group lasso problem (8) and fixes the NTK.

**UCI datasets.** We compare the regularied NTK with our IRLS algorithm (Algorithm 1) on the UCI ML Repository datasets. We follow the procedure described in [43] for $n \leq 1000$ to extract and standardize the datasets. We observe that our method achieves higher (or the same) test accuracy for 26 **out of** 33 datasets (see Table 1 for details) while the NTK achieves higher (or the same) test accuracy for 14 datasets which empirically supports our main claim that the IRLS procedure fixes the NTK. Details of the experiments can be found in Section B of the supplementary material.

## 8 Discussion and Limitations

In this work, we explored the connection between finite-width theories of neural networks given by convex reformulations and infinite-width theories of neural networks given by the NTK. To bridge these theories, we first interpreted the group lasso convex formulation of the gated ReLU network as a multiple kernel learning model using the masking kernels generated by its gates. Then, we linked this MKL model with the NTK of the gated ReLU network evaluated on the training data. Specifically, we showed that the NTK is equivalent to the weighted masking kernel with weights that depend only on the input data $\mathbf{X}$ and not on the targets $\mathbf{y}$. We contrast this with the MKL interpretation of the gated ReLU network which learns the optimal data-dependent kernel using both $\mathbf{X}$ and $\mathbf{y}$. Therefore, the NTK cannot perform better than the optimal MKL kernel. To fix the NTK, we improve the weights induced by it using the iteratively reweighted least squares (IRLS) scheme to obtain the optimal solution of the group lasso formulation of the gated ReLU network. We corroborated our theoretical results by empirically running IRLS on toy datasets.

While our theory is able to link the optimization properties of the NTK with those of finite width networks via the MKL characterization of group lasso, we do not derive explicit generalization results on the test set. Applying existing generalization theory for kernel methods [44, 45] to the MKL interpretation of the convex reformulation could be a promising direction for future work.

Finally, although we studied fully connected networks in this paper, our approach can be directly extended to various neural network architectures, e.g., threshold/binary networks [46], convolution networks [47], generative adversarial networks [48], NNs with batch normalization [49], autoregressive models [50], and Transformers [51, 52].

## Acknowledgements

This work was supported in part by the National Science Foundation (NSF) CAREER Award under Grant CCF-2236829, Grant DMS-2134248 and Grant ECCS-2037304; in part by the U.S. Army Research Office Early Career Award under Grant W911NF-21-1-0242; in part by the Stanford Precourt Institute; and in part by the ACCESS—AI Chip Center for Emerging Smart Systems through InnoHK, Hong Kong, SAR.

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

# Supplementary Material

## Table of Contents

## A  Proofs and Theoretical Results

### A.1  Proof of Lemma 2.2

The gradient feature map of (9) is

$$
\nabla_{\boldsymbol{\theta}} \tilde{f}_{\mathcal{G}}\left(\mathbf{x}, \boldsymbol{\theta}\right) = \frac{1}{\sqrt{2m}}
\begin{bmatrix}
\mathbb{1}\left\{\mathbf{x}^T \mathbf{g}_1 \geq 0\right\} \mathbf{x}^T \mathbf{w}_1^{(1)} \\
\vdots \\
\mathbb{1}\left\{\mathbf{x}^T \mathbf{g}_m \geq 0\right\} \mathbf{x}^T \mathbf{w}_m^{(1)} \\
\mathbb{1}\left\{\mathbf{x}^T \mathbf{g}_1 \geq 0\right\} w_1^{(2)} \mathbf{x} \\
\vdots \\
\mathbb{1}\left\{\mathbf{x}^T \mathbf{g}_m \geq 0\right\} w_m^{(2)} \mathbf{x}
\end{bmatrix}.
$$

The finite width random NTK given by $H_{m,\boldsymbol{\theta},\mathcal{G}}\left(\mathbf{x}, \mathbf{x}'\right) := \nabla_{\boldsymbol{\theta}} \tilde{f}_{\mathcal{G}}\left(\mathbf{x}, \boldsymbol{\theta}\right)^T \nabla_{\boldsymbol{\theta}} \tilde{f}_{\mathcal{G}}\left(\mathbf{x}', \boldsymbol{\theta}\right)$ is

$$
H_{m,\boldsymbol{\theta},\mathcal{G}}\left(\mathbf{x}, \mathbf{x}'\right) = \frac{1}{2m} \sum_{j=1}^{m} \left(\mathbb{1}\left\{\mathbf{x}^T \mathbf{g}_j \geq 0\right\} \mathbb{1}\left\{\mathbf{x}'^T \mathbf{g}_j \geq 0\right\} \left(\mathbf{x}^T \mathbf{w}_j^{(1)} \mathbf{x}'^T \mathbf{w}_j^{(1)} + \left(w_j^{(2)}\right)^2 \mathbf{x}^T \mathbf{x}'\right)\right).
$$

By the Law of Large numbers since $\mathbf{g}_j, \mathbf{w}_j^{(1)}, w_j^{(2)}$ are iid, $H_{m,\boldsymbol{\theta},\mathcal{G}}\left(\mathbf{x}, \mathbf{x}'\right)$ converges in probability to the expectation of the quantity in the sum as $m \to \infty$. We denote this infinite limit by $H\left(\mathbf{x}, \mathbf{x}'\right)$. Therefore,

$$
H\left(\mathbf{x}, \mathbf{x}'\right) = \frac{1}{2}\mathbb{E}\left[\mathbb{1}\left\{\mathbf{x}^T \mathbf{g} \geq 0\right\} \mathbb{1}\left\{\mathbf{x}'^T \mathbf{g} \geq 0\right\} \left(\mathbf{x}^T \mathbf{w}^{(1)} \mathbf{x}'^T \mathbf{w}^{(1)} + \left(w^{(2)}\right)^2 \mathbf{x}^T \mathbf{x}'\right)\right],
$$

where $\mathbf{g} \sim \mathcal{N}\left(\mathbf{0}, \mathbf{I_d}\right)$, $\mathbf{w}^{(1)} \sim \mathcal{N}\left(\mathbf{0}, \mathbf{I_d}\right)$, $w^{(2)} \sim \mathcal{N}\left(0, 1\right)$. Using $\mathbb{E}\left[\left(w^{(2)}\right)^2\right] = 1$, $\mathbb{E}\left[\mathbf{x}^T \mathbf{w}^{(1)} \mathbf{x}'^T \mathbf{w}^{(1)}\right] = \mathbf{x}^T \mathbf{x}'$ and the linearity of expectation, we get

$$
H\left(\mathbf{x}, \mathbf{x}'\right) = \mathbf{x}^T \mathbf{x}' \cdot \mathbb{E}\left[\mathbb{1}\left\{\mathbf{x}^T \mathbf{g} \geq 0\right\} \mathbb{1}\left\{\mathbf{x}'^T \mathbf{g} \geq 0\right\}\right]. \tag{18}
$$

To evaluate the expectation of the product of the indicator random variables, we condition on one of the indicators being equal to 1.

$$\mathbb{E}\left[\mathbb{1}\left\{\mathbf{x}^T\mathbf{g} \geq 0\right\}\mathbb{1}\left\{\mathbf{x}'^T\mathbf{g} \geq 0\right\}\right] = \mathbb{E}\left[\mathbb{1}\left\{\mathbf{x}^T\mathbf{g} \geq 0\right\} \mid \mathbb{1}\left\{\mathbf{x}'^T\mathbf{g} \geq 0\right\} = 1\right] \cdot \mathbb{P}\left[\mathbb{1}\left\{\mathbf{x}'^T\mathbf{g} \geq 0\right\} = 1\right]$$

$$= \frac{1}{2} \cdot \mathbb{E}\left[\mathbb{1}\left\{\mathbf{x}^T\mathbf{g} \geq 0\right\} \mid \mathbb{1}\left\{\mathbf{x}'^T\mathbf{g} \geq 0\right\} = 1\right].$$

The conditional expectation is the probability of the event that $\mathbf{x}^T\mathbf{g} \geq 0$ given that $\mathbf{x}'^T\mathbf{g} \geq 0$. This can be geometrically seen as the ratio of $\pi$ minus the angle between $\mathbf{x}$ and $\mathbf{x}'$ to $\pi$. So, we get

$$\mathbb{E}\left[\mathbb{1}\left\{\mathbf{x}^T\mathbf{g} \geq 0\right\}\mathbb{1}\left\{\mathbf{x}'^T\mathbf{g} \geq 0\right\}\right] = \frac{1}{2} \cdot \frac{1}{\pi}\left(\pi - \arccos\left(\frac{\mathbf{x}^T\mathbf{x}'}{\|\mathbf{x}\|_2\|\mathbf{x}'\|_2}\right)\right).$$

Combining this with (18) yields the desired result.

## A.2 Proof of Lemma 2.3

The gradient feature map of (11) is

$$\nabla_{\boldsymbol{\theta}} f_r\left(\mathbf{x}, \boldsymbol{\theta}\right) = \frac{1}{\sqrt{2m}}\begin{bmatrix} \mathbb{1}\left\{\mathbf{x}^T\mathbf{w}_1^{(+)} \geq 0\right\}\mathbf{x} \\ \vdots \\ \mathbb{1}\left\{\mathbf{x}^T\mathbf{w}_m^{(+)} \geq 0\right\}\mathbf{x} \\ -\mathbb{1}\left\{\mathbf{x}^T\mathbf{w}_1^{(-)} \geq 0\right\}\mathbf{x} \\ \vdots \\ -\mathbb{1}\left\{\mathbf{x}^T\mathbf{w}_m^{(-)} \geq 0\right\}\mathbf{x} \end{bmatrix}.$$

The finite width random NTK given by $H_{m,\boldsymbol{\theta}}^r\left(\mathbf{x}, \mathbf{x}'\right) := \nabla_{\boldsymbol{\theta}} f_r\left(\mathbf{x}, \boldsymbol{\theta}\right)^T \nabla_{\boldsymbol{\theta}} f_r\left(\mathbf{x}', \boldsymbol{\theta}\right)$ is

$$H_{m,\boldsymbol{\theta}}^r\left(\mathbf{x}, \mathbf{x}'\right) = \frac{1}{2m}\sum_{j=1}^m \left(\mathbb{1}\left\{\mathbf{x}^T\mathbf{w}_j^{(+)} \geq 0\right\}\mathbb{1}\left\{\mathbf{x}'^T\mathbf{w}_j^{(+)} \geq 0\right\}\mathbf{x}^T\mathbf{x}' + \right.$$

$$\left.\mathbb{1}\left\{\mathbf{x}^T\mathbf{w}_j^{(-)} \geq 0\right\}\mathbb{1}\left\{\mathbf{x}'^T\mathbf{w}_j^{(-)} \geq 0\right\}\mathbf{x}^T\mathbf{x}'\right).$$

By the Law of Large numbers since $\mathbf{w}_j^{(+)}, \mathbf{w}_j^{(-)}$ are iid, $H_{m,\boldsymbol{\theta}}^r\left(\mathbf{x}, \mathbf{x}'\right)$ converges in probability to the expectation of the quantity in the sum as $m \to \infty$. We denote this infinite limit by $H^r\left(\mathbf{x}, \mathbf{x}'\right)$. Therefore,

$$H^r\left(\mathbf{x}, \mathbf{x}'\right) = \frac{1}{2}\mathbb{E}\left[\mathbb{1}\left\{\mathbf{x}^T\mathbf{w}^{(+)} \geq 0\right\}\mathbb{1}\left\{\mathbf{x}'^T\mathbf{w}^{(+)} \geq 0\right\} + \right.$$

$$\left.\mathbb{1}\left\{\mathbf{x}^T\mathbf{w}^{(-)} \geq 0\right\}\mathbb{1}\left\{\mathbf{x}'^T\mathbf{w}^{(-)} \geq 0\right\}\right]\mathbf{x}^T\mathbf{x}',$$

where $\mathbf{w}^{(+)} \sim \mathcal{N}\left(\mathbf{0}, \mathbf{I_d}\right)$ and $\mathbf{w}^{(-)} \sim \mathcal{N}\left(\mathbf{0}, \mathbf{I_d}\right)$. Since $\mathbf{w}^{(+)}$ and $\mathbf{w}^{(-)}$ are i.i.d., we can combine the expectations to get

$$H^r\left(\mathbf{x}, \mathbf{x}'\right) = \mathbb{E}\left[\mathbb{1}\left\{\mathbf{x}^T\mathbf{w}^{(+)} \geq 0\right\}\mathbb{1}\left\{\mathbf{x}'^T\mathbf{w}^{(+)} \geq 0\right\}\right]\mathbf{x}^T\mathbf{x}'.$$

Comparing to (18), we get the desired result.

## A.3 Proof of Lemma 4.2

Since $\mathcal{G}$ is complete, $\forall\, i \in [p]\,,\exists\, \mathbf{g}_i \in \mathcal{G}$ such that $\mathrm{diag}\left(\mathbb{1}\left\{\mathbf{X}\mathbf{g}_i \geq 0\right\}\right) = \mathbf{D}_i$. Additionally, these gates $\mathbf{g}_i$ are unique since $\mathrm{diag}\left(\mathbb{1}\left\{\mathbf{X}\mathbf{g}_i \geq 0\right\}\right) \neq \mathrm{diag}\left(\mathbb{1}\left\{\mathbf{X}\mathbf{g}_j \geq 0\right\}\right) \implies \mathbf{g}_i \neq \mathbf{g}_j$. Therefore, there is a subset of gates $\mathcal{G}$ of size $p$ that are uniquely associated to each mask $\mathbf{D}_i$. Since $\mathcal{G}$ is also minimally complete, this subset must be $\mathcal{G}$ and this proves the lemma.

## A.4 Proof of Theorem 4.3

By Lemma 3.1, the MKL problem given by

$$\min_{\boldsymbol{\eta} \in \Delta_p, \mathbf{v}_i \in \mathbb{R}^d} \quad \left\| \sum_{i=1}^{p} \sqrt{\eta_i} \mathbf{D}_i \mathbf{X} \mathbf{v}_i - \mathbf{y} \right\|_2^2 + \hat{\lambda} \sum_{i=1}^{p} \|\mathbf{v}_i\|_2^2.$$

is equivalent to the group lasso problem (14). Additionally, (14) is also equivalent to the standard group lasso (15) since they have the same regularization paths. Since $\mathcal{G}$ is minimally complete, (7) (which is equivalent to (6) in the minimally complete gate set setting) is equivalent to the group lasso problem (15) and this proves the theorem.

## A.5 Proof of Theorem 5.1

The weighted masking kernel is given by

$$\mathbf{K}_{\mathcal{G}} (\tilde{\boldsymbol{\eta}}) = \sum_{i=1}^{p} \tilde{\eta}_i \mathbf{D}_i \mathbf{X} \mathbf{X}^T \mathbf{D}_i. \tag{19}$$

Now, the NTK matrix $\mathbf{H}$ can be written as (using (18)),

$$\mathbf{H} = \mathbb{E}_{\mathbf{g} \sim \mathcal{N}(\mathbf{0}, \mathbf{I_d})} \left[ \operatorname{diag} \left( \mathbb{1} \left\{ \mathbf{X} \mathbf{g} \geq \mathbf{0} \right\} \right) \mathbf{X} \mathbf{X}^T \operatorname{diag} \left( \mathbb{1} \left\{ \mathbf{X} \mathbf{g} \geq \mathbf{0} \right\} \right) \right].$$

We can condition on the events that $\operatorname{diag} \left( \mathbb{1} \left\{ \mathbf{X} \mathbf{g} \geq \mathbf{0} \right\} \right) = \mathbf{D}_i \, \forall \, i \in [p]$. Since $\mathcal{D}_{\mathbf{X}}$ covers all possible data maskings $\mathbf{D}_i$, we know that $\sum_{i=1}^{p} \mathbb{P} \left[ \operatorname{diag} \left( \mathbb{1} \left\{ \mathbf{X} \mathbf{g} \geq \mathbf{0} \right\} \right) = \mathbf{D}_i \right] = 1$. Therefore,

$$\mathbf{H} = \sum_{i=1}^{p} \mathbb{E}_{\mathbf{g} \sim \mathcal{N}(\mathbf{0}, \mathbf{I_d})} \left[ \mathbf{D}_i \mathbf{X} \mathbf{X}^T \mathbf{D}_i \mid \operatorname{diag} \left( \mathbb{1} \left\{ \mathbf{X} \mathbf{g} \geq \mathbf{0} \right\} \right) = \mathbf{D}_i \right] \mathbb{P} \left[ \operatorname{diag} \left( \mathbb{1} \left\{ \mathbf{X} \mathbf{g} \geq \mathbf{0} \right\} \right) = \mathbf{D}_i \right]$$

$$= \sum_{i=1}^{p} \mathbf{D}_i \mathbf{X} \mathbf{X}^T \mathbf{D}_i \mathbb{P} \left[ \operatorname{diag} \left( \mathbb{1} \left\{ \mathbf{X} \mathbf{g} \geq \mathbf{0} \right\} \right) = \mathbf{D}_i \right].$$

Plugging in $\tilde{\eta}_i = \mathbb{P} \left[ \operatorname{diag} \left( \mathbb{1} \left\{ \mathbf{X} \mathbf{g} \geq \mathbf{0} \right\} \right) = \mathbf{D}_i \right]$ and comparing to (19), we get that $\mathbf{K}_{\mathcal{G}} (\tilde{\boldsymbol{\eta}}) = \mathbf{H}$ and this proves Theorem 5.1.

## A.6 Proof of Corollary 5.2

First, we define a single combined masked data matrix $\tilde{\mathbf{X}} \in \mathbb{R}^{n \times pd}$ as follows

$$\tilde{\mathbf{X}} := [\mathbf{D}_1 \mathbf{X} \ \ \mathbf{D}_2 \mathbf{X} \cdots \mathbf{D}_p \mathbf{X}].$$

We also define a diagonal regularization weighting matrix $\mathbf{R} \in \mathbb{R}^{pd \times pd}$ as follows

$$\mathbf{R} := \operatorname{diag} \left( [\tilde{\eta}_1 \mathbf{1_d} \quad \tilde{\eta}_2 \mathbf{1_d} \quad \cdots \quad \tilde{\eta}_p \mathbf{1_d}] \right),$$

where $\mathbf{1_d}$ is a $d$-dimensional all-1s row vector. Now, we can rewrite (16) with $\boldsymbol{\Phi}_i = \mathbf{D}_i \mathbf{X}$ and regularization weights $\tilde{\boldsymbol{\eta}}$ as

$$\min_{\mathbf{w} \in \mathbb{R}^{pd}} \quad \left\| \tilde{\mathbf{X}} \mathbf{w} - \mathbf{y} \right\|_2^2 + \lambda \left\| \mathbf{R}^{-\frac{1}{2}} \mathbf{w} \right\|_2^2. \tag{20}$$

Setting the gradient of the convex objective in (20) to 0, we get the following equation for the solution $\tilde{\mathbf{w}}$,

$$2 \tilde{\mathbf{X}}^T \left( \tilde{\mathbf{X}} \tilde{\mathbf{w}} - \mathbf{y} \right) + 2 \lambda \mathbf{R}^{-1} \tilde{\mathbf{w}} = 0.$$

Solving for $\tilde{\mathbf{w}}$, we get

$$\tilde{\mathbf{w}} = \left(\tilde{\mathbf{X}}^T\tilde{\mathbf{X}} + \lambda\mathbf{R}^{-1}\right)^{-1}\tilde{\mathbf{X}}^T\mathbf{y}.$$

By applying the matrix inversion lemma [53], we can rewrite this as

$$\tilde{\mathbf{w}} = \mathbf{R}\tilde{\mathbf{X}}^T\left(\tilde{\mathbf{X}}\mathbf{R}\tilde{\mathbf{X}}^T + \lambda\mathbf{I_n}\right)^{-1}\mathbf{y}.$$

Now, we notice that

$$\sum_{i=1}^{p}\mathbf{D}_i\mathbf{X}\tilde{\mathbf{w}}_i = \tilde{\mathbf{X}}\tilde{\mathbf{w}} = \tilde{\mathbf{X}}\mathbf{R}\tilde{\mathbf{X}}^T\left(\tilde{\mathbf{X}}\mathbf{R}\tilde{\mathbf{X}}^T + \lambda\mathbf{I_n}\right)^{-1}\mathbf{y}. \tag{21}$$

Additionally, observe that $\tilde{\mathbf{X}}\mathbf{R}\tilde{\mathbf{X}}^T = \sum_{i=1}^{p}\tilde{\eta}_i\mathbf{D}_i\mathbf{X}\mathbf{X}\mathbf{D}_i = \mathbf{H}$ where the last equality is by Theorem 5.1. Plugging this into (21), we get

$$\sum_{i=1}^{p}\mathbf{D}_i\mathbf{X}\tilde{\mathbf{w}}_i = \mathbf{H}\left(\mathbf{H} + \lambda\mathbf{I_n}\right)^{-1}\mathbf{y}$$

and this proves Corollory 5.2.

### A.7   Proof of Theorem 6.1

We begin with the optimality of $\hat{\mathbf{w}}$,

$$\frac{1}{n}\left\|\sum_{i=1}^{p}\mathbf{D}_i\mathbf{X}\hat{\mathbf{w}}_i - \mathbf{y}\right\|_2^2 + \lambda\sum_{i=1}^{p}\|\hat{\mathbf{w}}_i\|_2 \leq \frac{1}{n}\left\|\sum_{i=1}^{p}\mathbf{D}_i\mathbf{X}\mathbf{w}_i^* - \mathbf{y}\right\|_2^2 + \lambda\sum_{i=1}^{p}\|\mathbf{w}_i^*\|_2.$$

By plugging in the model $\mathbf{y} = \sum_{i=1}^{p}\mathbf{D}_i\mathbf{X}\mathbf{w}_i^* + \epsilon$, we simplify this to

$$\frac{1}{n}\left\|\sum_{i=1}^{p}\mathbf{D}_i\mathbf{X}\left(\hat{\mathbf{w}}_i - \mathbf{w}_i^*\right) - \epsilon\right\|_2^2 + \lambda\sum_{i=1}^{p}\|\hat{\mathbf{w}}_i\|_2 \leq \frac{1}{n}\|\epsilon\|_2^2 + \lambda\sum_{i=1}^{p}\|\mathbf{w}_i^*\|_2.$$

We expand the square on the LHS and obtain the following bound on the prediction error

$$\frac{1}{n}\left\|\sum_{i=1}^{p}\mathbf{D}_i\mathbf{X}\left(\hat{\mathbf{w}}_i - \mathbf{w}_i^*\right)\right\|_2^2 \leq \frac{2}{n}\sum_{i=1}^{p}\epsilon^T\mathbf{D}_i\mathbf{X}\left(\hat{\mathbf{w}}_i - \mathbf{w}_i^*\right) + \lambda\sum_{i=1}^{p}\left(\|\mathbf{w}_i^*\|_2 - \|\hat{\mathbf{w}}_i\|_2\right)$$

$$\leq \frac{2}{n}\sum_{i=1}^{p}\left\|\epsilon^T\mathbf{D}_i\mathbf{X}\right\|_2\|\hat{\mathbf{w}}_i - \mathbf{w}_i^*\|_2 + \lambda\sum_{i=1}^{p}\left(\|\mathbf{w}_i^*\|_2 - \|\hat{\mathbf{w}}_i\|_2\right),$$

where we used Cauchy-Schwarz inequality in the second step. Now, since $\mathbf{D}_i$ are diagonal matrices with either $0$ or $1$ on the diagonals, we can immediately say that $\left\|\epsilon^T\mathbf{D}_i\mathbf{X}\right\|_2 \leq \left\|\epsilon^T\mathbf{X}\right\|_2$ for all $i = 1, \ldots, p$. As a consequence, we get

$$\frac{1}{n}\left\|\sum_{i=1}^{p}\mathbf{D}_i\mathbf{X}\left(\hat{\mathbf{w}}_i - \mathbf{w}_i^*\right)\right\|_2^2 \leq \frac{2}{n}\left\|\epsilon^T\mathbf{X}\right\|_2\sum_{i=1}^{p}\|\hat{\mathbf{w}}_i - \mathbf{w}_i^*\|_2 + \lambda\sum_{i=1}^{p}\left(\|\mathbf{w}_i^*\|_2 - \|\hat{\mathbf{w}}_i\|_2\right)$$

$$\leq \left(\frac{2}{n}\left\|\epsilon^T\mathbf{X}\right\|_2 + \lambda\right)\sum_{i=1}^{p}\|\mathbf{w}_i^*\|_2 + \left(\frac{2}{n}\left\|\epsilon^T\mathbf{X}\right\|_2 - \lambda\right)\sum_{i=1}^{p}\|\hat{\mathbf{w}}_i\|_2,$$

where we used the triangle inequality in the second step. Now, if $\lambda \geq \frac{2}{n}\left\|\epsilon^T\mathbf{X}\right\|_2$, we can get rid of the second term, namely $\left(\frac{2}{n}\left\|\epsilon^T\mathbf{X}\right\|_2 - \lambda\right)\sum_{i=1}^{p}\|\hat{\mathbf{w}}_i\|_2$ and obtain a bound on the prediction error that does not depend on $\hat{\mathbf{w}}$. To this end, notice that $\epsilon^T\mathbf{X}$ is a multivariate Gaussian with mean $\mathbf{0}$ and covariance $\sigma^2\mathbf{X}^T\mathbf{X}$. We now present a standard concentration result for the norms of Gaussians in the lemma below,

**Lemma A.1.** *Let $\mathbf{z}$ be a $d$-dimensional multivariate Gaussian with mean $\mathbf{0}$ and covariance matrix $\mathbf{\Sigma}$. We have*

$$\mathbb{P}\left[\|\mathbf{z}\|_2 \le z\right] \ge 1 - 2\exp\left(\frac{-z^2}{2\operatorname{tr}(\mathbf{\Sigma})}\right).$$

Applying Lemma A.1 to $\epsilon^T \mathbf{X}$, we get

$$\mathbb{P}\left[\left\|\epsilon^T \mathbf{X}\right\|_2 \le z\right] \ge 1 - 2e^{\frac{-z^2}{2\sigma^2 \operatorname{tr}(\mathbf{X}^T\mathbf{X})}}.$$

Plugging in $z = \frac{n}{2}\lambda$ where $\lambda = \frac{t\sigma\sqrt{\operatorname{tr}(\mathbf{X}^T\mathbf{X})}}{n}$ as in the statement of theorem, we get

$$\mathbb{P}\left[\frac{2}{n}\left\|\epsilon^T\mathbf{X}\right\|_2 \le \lambda\right] \ge 1 - 2e^{\frac{-t^2}{8}}.$$

Therefore, by choosing $\lambda = \frac{t\sigma\sqrt{\operatorname{tr}(\mathbf{X}^T\mathbf{X})}}{n}$, we have that $\frac{2}{n}\left\|\epsilon^T\mathbf{X}\right\|_2 \le \lambda$ with probability at least $1 - 2e^{-t^2/8}$. Consequently, we have

$$\frac{1}{n}\left\|\sum_{i=1}^{p}\mathbf{D}_i\mathbf{X}\left(\hat{\mathbf{w}}_i - \mathbf{w}_i^*\right)\right\|_2^2 \le 2\lambda\sum_{i=1}^{p}\|\mathbf{w}_i^*\|_2$$

with probability at least $1 - 2e^{-t^2/8}$, which completes the proof.

### A.7.1 Proof of Lemma A.1

Since the norm of Gaussian vector does not change by applying a rotation, we know that $\left\|\Lambda^{1/2}\mathbf{u}\right\|_2$ and $\|\mathbf{z}\|_2$ have the same distribution where $\Lambda$ is the matrix of eigenvalues from the spectral decomposition $\mathbf{\Sigma} = Q\Lambda Q^T$ and $\mathbf{u} \sim \mathcal{N}(\mathbf{0}, \mathbf{I}_d)$. Now we apply a standard Chernoff bound argument. We have for some $s > 0$,

$$\begin{aligned}
\mathbb{P}\left[\left\|\Lambda^{1/2}\mathbf{u}\right\|_2 > z\right] &\le \mathbb{P}\left[\left\|\Lambda^{1/2}\mathbf{u}\right\|_1 > z\right]\\
&\le \mathbb{P}\left[e^{s\left\|\Lambda^{1/2}\mathbf{u}\right\|_1} > e^{sz}\right]\\
&\le e^{-sz}\Pi_{i=1}^{d}\mathbb{E}\left[e^{s\sqrt{\lambda_i}|\mathbf{u}_i|}\right]\\
&\le e^{-sz}\Pi_{i=1}^{d}2e^{s^2\lambda_i/2}.
\end{aligned}$$

By optimizing the bound over $s$, we get

$$\mathbb{P}\left[\left\|\Lambda^{1/2}\mathbf{u}\right\|_2 > z\right] \le 2\exp\left(\frac{-z^2}{2\sum_{i=1}^{d}\lambda_i}\right) \le 2\exp\left(\frac{-z^2}{2\operatorname{tr}(\mathbf{\Sigma})}\right).$$

Finally, using $\mathbb{P}\left[\|\mathbf{z}\|_2 \le z\right] = 1 - \mathbb{P}\left[\|\mathbf{z}\|_2 > z\right]$ gives the result.

## B Experimental Details

**Hyperplane Arrangements for 1D example.** In the 1D example in Figure 1, the data matrix $\mathbf{X}$ is in $\mathbb{R}^{n\times 2}$, i.e. the dimension $d = 2$. This is because we set the second component of each input to be exactly equal to 1 to simulate a bias in the neural network. Though the datapoints lie in $\mathbb{R}^2$, they have some structure which allows us to enumerate all hyperplane arrangements $\mathcal{D}_\mathbf{X}$ easily. Specifically, the datapoints all lie on the line $y = 1$ in the $xy$ plane, so the hyperplanes that separate them always separate them into two contiguous segments. Therefore, by sorting the datapoints according to their first component, we can directly enumerate all $2n$ hyperplane arrangements.

**Nonconvex ReLU network.** For the black dashed line in Figure 1, we trained a two layer ReLU network with 100 hidden neurons using the standard weight decay regularized objective with gradient descent using a learning rate of 0.01 for 200000 epochs.

**Convex program solver specifications.** For directly solving the group lasso problem (8), we used CVXPY with the MOSEK solver [54]. For the $\ell_2$ regularized least squares problems (16), we used iterative methods with an efficient matrix-vector product implementation to avoid explicitly constructing the large combined masked data matrix $\tilde{\mathbf{X}}$. Specifically, we used the LSQR and LSMR methods as described in [41, 42]. The experiments were run locally on a MacBook Air 2020 version with the Apple M1 chip.

**UCI Machine Learning Repository.** Here we present details for the experiments on several real datasets from the UCI Machine Learning Repository [55]. We follow the procedure described in [43] for $n \leq 1000$ to extract and standardize the datasets. For these experiments, we use the $75\% - 25\%$ ratio for the training and test set splits. We use the squared loss and tune the regularization parameter $\lambda$ for both NTK and our approach by performing a grid search over the set $\{10^{-5}, 10^{-4}, 10^{-3}, 10^{-2}, 10^{-1}, 10^0, 10^1\}$. As shown in Table 1, our method achieves higher (or the same) test accuracy for 26 datasets out 33 datasets while standard NTK achieves higher (or the same) test accuracy for 14.

## C  Generic Loss Functions

Here, we prove that the analysis with squared loss presented in the main paper can be extended to arbitrary convex loss function. Let us consider the following regularized two-layer ReLU network training problem with an arbitrary convex loss function $\mathcal{L}(\cdot, \cdot)$

$$\min_{\mathbf{W}^{(1)}, \mathbf{w}^{(2)}} \mathcal{L}\left(\sum_{j=1}^m \left(\mathbf{X}\mathbf{w}_j^{(1)}\right)_+ w_j^{(2)}, \mathbf{y}\right) + \lambda \sum_{j=1}^m \left(\left\|\mathbf{w}_j^{(1)}\right\|_2^2 + |w^{(2)}|^2\right). \tag{22}$$

Then, as shown in [7], the corresponding gated convex reformulation of (22) is

$$P^* := \min_{\mathbf{w}_i} \mathcal{L}\left(\sum_{i=1}^p \mathbf{D}_i \mathbf{X} \mathbf{w}_i - \mathbf{y}\right) + \lambda \sum_{i=1}^p \|\mathbf{w}_i\|_2. \tag{23}$$

In order to realize the emergence of kernel characterization in the case of arbitrary loss function, we use Lagrange duality. Particularly, since (23) is a convex optimization problem and Slater's conditions holds as proven in [7, 8], we equivalently represent (22) as the following dual problem

$$P^* = \max_{\mathbf{z}} -\mathcal{L}^*(\mathbf{z}, \mathbf{y}) \quad \text{s.t.} \quad \left\|\mathbf{z}^T \mathbf{D}_i \mathbf{X}\right\|_2 \leq \lambda, \forall i \in [p] \tag{24}$$

where $\mathcal{L}(\mathbf{z}, \mathbf{y})$ is the Fenchel conjugate of $\mathcal{L}(\cdot, \mathbf{y})$ defined as [56]

$$\mathcal{L}^*(\mathbf{z}, \mathbf{y}) := \max_{\mathbf{v}} \mathbf{v}^T \mathbf{z} - \mathcal{L}(\mathbf{v}, \mathbf{y}).$$

We then form the Lagrangian for (24) as follows

$$L(\mathbf{z}, \alpha_i) = -\mathcal{L}^*(\mathbf{z}, \mathbf{y}) + \sum_{i=1}^p \alpha_i \left(\lambda - \left\|\mathbf{z}^T \mathbf{D}_i \mathbf{X}\right\|_2\right)$$

$$= -\mathcal{L}^*(\mathbf{z}, \mathbf{y}) + \sum_{i=1}^p \alpha_i \left(\lambda - \mathbf{z}^T \mathbf{D}_i \mathbf{X} \mathbf{X}^T \mathbf{D}_i \mathbf{z}\right)$$

$$= -\mathcal{L}^*(\mathbf{z}, \mathbf{y}) + \sum_{i=1}^p \alpha_i \left(\lambda - \mathbf{z}^T \mathbf{K}_i \mathbf{z}\right),$$

where $\alpha_i \geq 0$ is the Lagrange multiplier for the inequality constraint. Based on the representation above, the dual problem in (24) can be written as

$$P^* = \max_{\mathbf{z}} \min_{\alpha_i \geq 0} -\mathcal{L}^*(\mathbf{v}, \mathbf{y}) + \sum_{i=1}^p \alpha_i \left(\lambda - \mathbf{z}^T \mathbf{K}_i \mathbf{z}\right). \tag{25}$$

Thus, similar to (12), the problem in (25) can be globally optimized with alternating minimization [33, 35].

# D  Extensions to Deeper Networks

In this section, we generalize the kernel characterization of two-layer ReLU networks presented in the main paper to deeper architectures. We first note that most prior works on convex reformulation of ReLU networks studied only two-layer ReLU networks, e.g., [6–10], since the non-convex interaction of multiple nonlinear layers hinders the convex analysis of deeper networks. However, recent follow-up studies [13, 57, 58] showed that the analysis can be extended to arbitrarily deep networks. Based on this recent results, below we briefly explain how one can extend our kernel characterization to three-layer networks. By following the convex reformulations in [57, 58], we have the following gated ReLU training problem for three-layer network

$$\min_{\mathbf{w}_{ij}} \left\| \sum_{i=1}^{p_1} \sum_{j=1}^{p_2} \mathbf{D}_i^{(1)} \mathbf{D}_j^{(2)} \mathbf{X} \mathbf{w}_{ij} - \mathbf{y} \right\|_2^2 + \lambda \sum_{i=1}^{p_1} \sum_{j=1}^{p_2} \|\mathbf{w}_{ij}\|_2, \tag{26}$$

where the hyperplane arrangements for the first and second ReLU layers, i.e., denoted as $\mathbf{D}_i^{(1)} \in \mathcal{D}_{\mathbf{X}}^{(1)}$ and $\mathbf{D}_j^{(2)} \in \mathcal{D}_{\mathbf{X}}^{(2)}$, are defined as follows

$$\mathcal{D}_{\mathbf{X}}^{(1)} := \left\{ \operatorname{diag}\left(\mathbb{1}\left\{\mathbf{X}\mathbf{u} \geq \mathbf{0}\right\}\right) : \mathbf{u} \in \mathbb{R}^d \right\} \text{ and } p_1 := \left| \mathcal{D}_{\mathbf{X}}^{(1)} \right| \tag{27}$$

$$\mathcal{D}_{\mathbf{X}}^{(2)} := \left\{ \operatorname{diag}\left(\mathbb{1}\left\{\left(\mathbf{X}\mathbf{U}^{(1)}\right)_+ \mathbf{u}^{(2)} \geq \mathbf{0}\right\}\right) : \mathbf{U}^{(1)} \in \mathbb{R}^{d \times m_1}, \mathbf{u}^{(2)} \in \mathbb{R}^{m_1} \right\} \text{ and } p_2 := \left| \mathcal{D}_{\mathbf{X}}^{(2)} \right|.$$

Then, following the derivations in Section 4, we can conveniently express the corresponding feature matrices of these masking feature maps on the data $\mathbf{X}$ in terms of fixed diagonal data masks as $\boldsymbol{\Phi}_{ij} = \mathbf{D}_i^{(1)} \mathbf{D}_j^{(2)} \mathbf{X}$. Similarly, the corresponding masking kernel matrices take the form $\mathbf{K}_{ij} = \mathbf{D}_i^{(1)} \mathbf{D}_j^{(2)} \mathbf{X} \mathbf{X}^T \mathbf{D}_i^{(1)} \mathbf{D}_j^{(2)}$. The same interpretation also extends to arbitrarily deep networks and yields a more complicated feature map due to the interaction of multiple hyperplane arrangement matrices, e.g., $\mathbf{D}_i^{(1)} \mathbf{D}_j^{(2)} \mathbf{D}_k^{(3)} \ldots$

# E  Cone Decomposition

Here, we briefly review the cone decomposition concept introduced by [8]. We first note that (3) is the exact convex reformulation of the standard non-convex regularized training problem in (2). However, as mentioned in the main paper, the number of constraints in (3) can be prohibitively large, precisely can be $\mathcal{O}(n^d)$, and consequently prevents solving the exact convex optimization problem especially for large scale datasets. To remedy this issue, [8] proposed the unconstrained formulation in (8), but, this relaxation corresponds to another neural network architecture called gated ReLU network in (5). Although this relaxation breaks dependence between the arrangements and preactivations of a ReLU activation, [8] proved that once the gated ReLU network training problem in (8) training problem is solved, one can construct a solution for the ReLU network model in (1). This procedure is named as *cone decomposition* and we provide details regarding the implementation steps below.

Let us assume that we solve the gated ReLU network training problem in (8) and denote the optimal solution as $\{\mathbf{w}_i^*\}_{i=1}^p$. Then, since each $\mathbf{D}_i \mathbf{X} \mathbf{w}_i^*$ can have both negative and positive entries (due to the absence of nonnegativity constraint ), we need to decompose $\mathbf{D}_i \mathbf{X} \mathbf{w}_i^*$ into two separate cones to maintain nonnegativity property of a ReLU activation. We formulate this decomposition step as the following convex optimization problem

$$\min_{\mathbf{v}_i, \mathbf{u}_i} \|\mathbf{v_i}\|_2 + \|\mathbf{u}_i\|_2 \quad \text{s.t.} \quad \mathbf{D}_i \mathbf{X} \mathbf{w}_i^* = \mathbf{D}_i \mathbf{X}(\mathbf{v}_i - \mathbf{u}_i), \quad \begin{matrix} (2\mathbf{D}_i - \mathbf{I}_n)\mathbf{X}\mathbf{v}_i \geq \mathbf{0} \\ (2\mathbf{D}_i - \mathbf{I}_n)\mathbf{X}\mathbf{u}_i \geq \mathbf{0} \end{matrix}. \tag{28}$$

After solving the convex optimization problem in (28) for all $i \in [p]$. We declare the optimal solutions, $\{\mathbf{v}_i^*, \mathbf{u}_i^*\}_{i=1}^p$ as a solution to the exact ReLU model in (1). Thus, instead of directly solving the convex reformulation with extremely large number of constraint in (3), we first solve the relaxed gated ReLU problem in (8) and then map the solutions to the proper cones to enforce the nonnegativity property of the ReLU activation.

### E.1 Computational Complexity

Although [8] reduce the computational complexity by eliminating the constraints in the convex optimization problem in (3), the number of variables, denotes as $p$, can still be exponentially large in feature dimension $d$. However, there are multiple ways to remedy this issue.

- We can change the architecture. Particularly, we can replace fully connected networks with convolutional networks. Then, since CNNs operate on the patch matrices $\{\mathbf{X}_b\}_{b=1}^{B}$ instead of the full data matrix $\mathbf{X}$, where $\mathbf{X}_b \in \mathbb{R}^{n \times h}$ and $h$ denotes the filter size, even when the data matrix is full rank, i.e., $r = \min(n, d)$, the number of hyperplane arrangements $p$ is upperbounded as $p \leq \mathcal{O}(n^{r_c})$, where $r_c := \max_b \operatorname{rank}(\mathbf{X}_b) \leq h \ll \min(n, d)$. For instance, let us consider a CNN with $3 \times 3$ filters, then $r_c \leq 9$ independent of $n, d$. As a consequence, weight sharing structure in CNNs dramatically limits the number of possible hyperplane arrangements and avoids exponential complexity. This also explains efficiency and remarkable generalization performance of CNNs in practice.

- We can also use a sampling based approach where one can randomly sample a tiny subset of all possible hyperplane arrangements and then solve the convex program with this subset. Thus, although the resulting approach is not exact, the training complexity will not be exponential in $d$ anymore. The experimental results in Appendix C show that this approximation in fact works extremely well, specifically resulting in models that outperform the NTK in 26/33 UCI datasets as detailed in Table 1.

