# OpenReview forum: "Fixing the NTK: From Neural Network Linearizations to Exact Convex Programs"
_NeurIPS.cc/2023/Conference — NeurIPS 2023 poster_

### Official Review · Reviewer_UwLS · 2023-07-05

**Soundness:** 3 good
**Presentation:** 3 good
**Contribution:** 3 good
**Rating:** 5
**Confidence:** 1

**Summary:**

The paper provides a connection of the convex program for gated ReLU networks to multiple kernel learning model with a weighted data masking feature map. Additionally, the paper provides a theoretical analysis of the predictive error of the proposed kernel algorithm.

**Strengths:**

The paper provides a new framework that employs multiple kernel learning which is a convex reformulation of neural network training process, the new framework is better able to explain the remarkable performance of neural networks than the neural tangent kernel perspective, which requires the infinite width limit assumption.

**Weaknesses:**

As I am not familiar with multiple kernel learning and group lasso, I feel that I am not able to fully identify the weakness in this paper. I will give a borderline rating with a low confidence score.

**Questions:**

No.

**Limitations:**

Yes.

---

### Official Review · Reviewer_nJAh · 2023-07-06

**Soundness:** 3 good
**Presentation:** 3 good
**Contribution:** 2 fair
**Rating:** 5
**Confidence:** 3

**Summary:**

This work considers a convex formulation of a finite-width regularised two layer ReLU network and interpretations as multiple kernel learning. This then is related to the neural tangent kernel.  The convex formulation considers cones of parameters with fixed activation along with a bound O(n^d). The gated ReLU further decouples the activation region from the linear function. A main conclusion is that while the NTK of the gated ReLU does not take labels into account, optimal Multiple Kernel Learning can.

**Strengths:**

* The work deals with a relevant problem of understanding the optimization problem of neural networks beyond the infinite width setting.
* The work gives an overview of different perspectives about ReLU networks, particularly previous work on convex programs, group lasso, multiple kernel learning, and NTK, which are interesting as they could facilitate further perspectives.
* The work discusses convex formulations of gated ReLU networks from a perspective of multiple kernel learning.

**Weaknesses:**

* The article indicates that previous works have considered training dynamics with infinitesimally small learning rates on networks in the infinite width limit. It would be indicated to comment on works covering small width [1,2], or works that discuss NTK more independently of the width [3], or works that investigate the dynamics of finite-width networks using NTK [4].
    * [1] E et al. A comparative analysis of optimization and generalization properties of two-layer neural network and random feature models under gradient descent dynamics.
    * [2] Su and Yang. On learning over-parameterized neural networks: A functional approximation perspective.
    * [3] Liu et al. On the linearity of large non-linear models: when and why the tangent kernel is constant.
    * [4] Bowman and Montufar. Spectral Bias Outside the Training Set for Deep Networks in the Kernel Regime.
* I missed a more explicit discussion of how the presented work is significantly different from or significantly advances in relation to prior works on convex formulations of ReLU networks, particularly the sequence of works [1,2,…].
    * [1] Ergen and Pilanci. Convex geometry and duality of over-parametrized neural networks.
    * [2] Pilanci and Ergen. Neural networks are convex regularizers: exact polynomial time convex optimization formulations for two layer networks.
* A more explicit connection between the presented analysis and the result of training a neural network would have made this a stronger contribution. Particularly, the conversion of gated ReLU to ReLU should be discussed more explicitly in the main part of the document. The theoretical discussion of fixing the kernel or improving the weights seems to focus on the training error with a more explicit discussion of generalization error missing. Figure 1 and the first experiments seem to be on toy examples. The experiments on UCI data compare NTK vs iteratively reweighed least squares but seem to be missing comparison with the trained network as well as reporting training accuracy.

**Questions:**

* Please clarify the intended notation for w^{(+)} and w^{(-)} or why (11) should have the same expressive power as (1). Concretely, (11) is a signed sum of 2m ReLUs, whereas (1) is a sum of just m ReLUs. For d=2 and m=1, (11) can represent functions which have 4 linear regions, whereas any function represented by (1) has at most 2 linear regions.

**Limitations:**

* The work seems to consider minimal complete gate sets.
* The work does not appear to include observations into overfitting or generalisation.

---

> ### Author Rebuttal · Authors · 2023-08-09
>
> We would like to thank the reviewer for the feedback and comments. We hope that you would consider increasing your score if your concerns are adequately addressed. We have addressed the issues related to generalization in the global response. Please see our responses to your other specific queries below.
>
> We greatly appreciate the reviewer for pointing out the references that study the NTK in the finite but large or small width regimes. We will include a discussion of this related work in the final revision of our work. However, we would like to point out an important distinction between our kernel characterization and the works which study the NTK in various width regimes. Specifically, based on our understanding, the works you reference still analyze the NTK in the regime where the weights do not much from initialization (the lazy regime), and the kernel is not learned from the data. On the other hand, our MKL kernel characterization does not require the weights to stay close to initialization, and the optimal kernel is learned from the data.
>
> The significant contribution in relation to prior works on convex formulations is to connect these reformulations to the MKL perspective, which further reveals a connection to the NTK theory, allowing us to show that the convex reformulation leads to a new kernel characterization of gated ReLU networks that applies beyond the lazy regime (which is required for the NTK theory to hold).
>
> $\textbf{Training accuracy results}$
> In the paper, we present experimental results that showcase the $\textit{generalization}$ performance of the gated ReLU model compared to the NTK. For completeness, we also present here the $\textit{training}$ accuracies for the UCI datasets, which we will add to the final revision of the paper. Note that, as a consequence of Theorems 4.3 and 5.1, we expect that our kernel always outperforms the NTK on the training set, which is exactly what we observe in the table below:
>
> **We note that our aim is to minimize the regularized training objective which includes not only the training loss but also the regularization term. And then during the validation process, we made a grid search for the regularization coefficient $\beta$ and included the results with the best test accuracies (generalization performance) for both models. Therefore, it is normal to have some training accuracy below 100%.**
>
> **Training accuracies by training on standard $\ell_2$ regularized loss**
>
> | Dataset Name                              | NTK | **Ours** |
> |-----------------------------------|---------|---------|
> | acute-inflammation                | 1.000   | 1.000   |
> | acute-nephritis                   | 1.000   | 1.000   |
> | balloons                          | 1.000   | 1.000   |
> | blood                             | 0.841   | 0.957   |
> | breast-cancer                     | 0.799   | 0.986   |
> | breast-cancer-wisc-prog           | 0.975   | 1.000   |
> | breast-cancer-wisc-diag           | 1.000   | 1.000   |
> | breast-cancer-wisc-prog           | 0.973   | 1.000   |
> | congressional-voting              | 0.752   | 0.791   |
> | conn-bench-sonar-mines-rocks      | 1.000   | 1.000   |
> | credit-approval                   | 0.868   | 1.000   |
> | cylinder-bands                    | 0.766   | 1.000   |
> | echocardiogram                    | 0.888   | 1.000   |
> | fertility                         | 0.960   | 0.920   |
> | haberman-survival                 | 0.821   | 0.991   |
> | heart-hungarian                   | 0.850   | 1.000   |
> | hepatitis                         | 0.931   | 1.000   |
> | ilpd-indian-liver                 | 0.808   | 1.000   |
> | ionosphere                        | 0.939   | 1.000   |
> | mammographic                      | 0.845   | 0.950   |
> | molec-biol-promoter               | 1.000   | 1.000   |
> | musk-1                            | 1.000   | 1.000   |
> | oocytes_trisopterus_nucleus_2f    | 0.830   | 1.000   |
> | parkinsons                        | 0.884   | 1.000   |
> | pima                              | 0.837   | 1.000   |
> | pittsburg-bridges-T-OR-D          | 0.908   | 0.987   |
> | planning                          | 0.809   | 1.000   |
> | statlog-australian-credit         | 0.571   | 1.000   |
> | statlog-german-credit             | 0.828   | 1.000   |
> | statlog-heart                     | 0.881   | 1.000   |
> | tic-tac-toe                       | 0.978   | 1.000   |
> | trains                            | 1.000   | 1.000   |
> | vertebral-column-2clases          | 0.892   | 1.000   |
>
> Our kernel achieves a higher train accuracy on 32/33 datasets, which is to be expected. The discrepancy in one of the datasets can be attributed to the approximation of subsampling a set of hyperplane arrangements.
>
>
>
>
>
> $\textbf{Clarification about reparameterized ReLU network}$
> We would thank the reviewer for bringing up this point. Indeed you are correct that the model (11) can be more expressive than (1), so we will modify the wording to "... can represent the same or more functions as (1) ...". Note that this does not affect the significance of our results, since the motivation behind introducing this reparameterization is to study an equally sized model (with $O(m)$ neurons) that can be obtained from a ReLU network (1) with $m$ neurons that has the same NTK as the gated ReLU network.
>
> $\textbf{Minimal gate sets}$
> Indeed for the theoretical results to go through, we require that the gate set be complete. However, in practice we observe that sampling gates to obtain a subset of all possible hyperplane arrangements performs very well, so the approach is computationally feasible. Additionally, as discussed in the global response, the number of arrangements $p$ can be significantly reduced by subsampling or using a CNN architecture.

---

> > ### Comment · Reviewer_nJAh · 2023-08-19
> > **Response to rebuttal**
> >
> > I appreciate the authors responses, which clarify some of the concerns in my initial review and propose suggestions to improve the text. I have adjusted my score accordingly.

---

> ### Author Response · Authors · 2023-08-19
> **Look forward to your feedback**
>
> Dear Reviewer nJAh,
>
> We believe that we have addressed your concerns in our responses. Since the deadline is approaching, we would like to hear your feedback so that we can respond to that before the discussion period ends. Please feel free to raise questions if you have other concerns. Thank you very much for your support, we really appreciate that!
>
> Best regards,
>
> Authors

---

### Official Review · Reviewer_Tkhd · 2023-07-08

**Soundness:** 3 good
**Presentation:** 3 good
**Contribution:** 3 good
**Rating:** 7
**Confidence:** 4

**Summary:**

The paper studies gated ReLU networks with L2 regularization. The authors show that this model is equivalent to Multiple Kernel Learning with group lasso, which is a convex optimzation problem. Thus L2 regularized gated ReLU networks is equivalent to learning the NTK according to a Lasso objective and then fitting the data with the resulting learned NTK. In contrast, in the NTK limit the model fits the data with the initial NTK, which is in general not optimal w.r.t. to the optimization procedure defined previously.

**Strengths:**

The article is well written and studies the important question of whether DNN can be interpreted as a kernel learning model. The theoretical results appear to be correct, and the numerical experiments illustrate the theory well.

**Weaknesses:**

The paper studies gated ReLU networks as an approximation of ReLU networks, but a lot of either very similar or stronger results to the ones proven here are already proven for actual ReLU networks. In particular, [Francis Bach, Breaking the curse of dimensionality with convex neural networks, 2017], proves a similar convex reformulation, and though not explicitely stated, it can also be interpreted as a kernel learning objective (arguably this optimization over kernels is more obvious in [A. Jacot, E. Golikov, C. Hongler, F. Gabriel, Feature Learning in L2-regularized DNNs: Attraction/Repulsion and Sparsity, NeurIPS 2022] which generalizes Francis Bach's results  to the deep case). Francis Bach also proves generalization bounds which are missing here.

The only possible advantage of the gated ReLU setting is that the optimization might not be NP-hard as is argued in the paper (we know the NP-hardness of the equivalent reformulation of ReLU networks), though this relies on the fact that the number of required gates $p$ grows polynomially, which is not proven. Actually I think it is quite likely that optimization proposed in this paper is NP-hard as well.

Another related problem is that the gated ReLU optimization is arguably `less unique' than the traditional ReLU, since the dates $g_i$ can always be changed without changing the activations on the training data, thus changing the learned function and kernel outside the training set. It appears that the solution proposed here is to arbitrarily choose a gate direction $g_i$ for each activation pattern on the training set, which is not ideal. Also this problem is not really addressed in the paper. Note that the same problem appear in ReLU networks though it seems less severe, since there can be case where you obtain uniqueness of the solution, in contrast it appears that there is never uniqueness with gated ReLU.

**Questions:**

I also do not like how the authors say in the abstract and introduction that the NTK at initialization is suboptimal compared to the learned NTK, without clarifying w.r.t. to which cost. This could suggest that they prove that the learned NTK is better in terms of generalization, which is not proven. Rather the learned NTK is optimal w.r.t. to the MKL optimization with group Lasso, which is a very specific optimization introduced in this paper (and previous paper) as a reformulation of the L2 regularization loss. So in some sense the statement that the learned NTK is optimal in contrast to the initial NTK is a tautology, since it is w.r.t. to a cost that was designed to reflect the effect of learning.

If we are instead talking about generalization, then we cannot say that the learned kernel is always better than the initial kernel, as there could be some form of `kernel overfitting'. This is the `no free lunch' theorem: there is no statistical model that is strictly better than another, each model is better under certain assumptions on the task. For example we already know that learning with a constant NTK is optimal if the true function $f^*$ is a Gaussian process with covariance equal to the limiting NTK, in such a setting, NTK learning would be detrimental. For these reasons, I ask the authors to be more specific when thay say that the learned kernel is optimal and initial NTK is suboptimal.

**Limitations:**

As already mentioned, the authors should be clearer about the two following possible issues:
- That the reformulation may not be that much faster, because the number of required kernels $d$ one has to optimize over could be very large. It is therefore not clear that gated ReLU offer a significant optimization advantage over ReLU networks.
- That the optimization is not unique outside the training set.

---

> ### Author Rebuttal · Authors · 2023-08-09
>
> We would like to thank the reviewer for the feedback and comments. We have addressed the issues related to complexity in the global response. Please see our responses to your other specific queries below.
>
> $\textbf{Regarding the objective}$
> We would like to clarify that one of the main results of our work is that the MKL optimization problem with group lasso is indeed equivalent to the standard non-convex training with L2 regularization loss. Thus, this cost was not designed from the perspective of the NTK, but instead was obtained by deriving an equivalent convex reformulation, which we also show captures the loss function from performing kernel ridge regression with the NTK.
>
> We thank the reviewer for raising this point, and we will revise our wording in the final version to be clearer when describing the suboptimality of the NTK.

---

> > ### Comment · Reviewer_Tkhd · 2023-08-10
> >
> > Optimizing over a polynomial a random subset of hyperplane alignement is an interesting idea and does yield a polynomial approximation to the full optimization,  but it does not answer the question of whether the convex reformulation of gated ReLU networks is computationaly more efficient than the convex reformulation of ReLU networks that I mentioned (since one could use a similar polynomial approximation of this second reformulation). In general I think it would be important to compare the method proposed here not only to the fixed NTK limit, but also to the other convex reformulations that other reviewers and I have mentioned, since gated ReLU are much less standard than ReLU networks, their use as a replacement for ReLU nets should be motivated.

---

> > > ### Author Response · Authors · 2023-08-15
> > >
> > > Thank you for your response. The main benefit of considering the gated ReLU network compared to the standard ReLU network is that the convex reformulation of the gated ReLU network leads to an _unconstrained_ optimization problem which is easier to solve than the standard ReLU network which leads to a cone constrained convex problem.

---

### Official Review · Reviewer_NJAL · 2023-07-10

**Soundness:** 4 excellent
**Presentation:** 4 excellent
**Contribution:** 4 excellent
**Rating:** 8
**Confidence:** 4

**Summary:**

This work presents the insight that the convex formulation of training a gated ReLU network is an instance of Multiple Kernel Learning (MKL) techniques. This contrasts with the NTK limit, which becomes a single kernel learning in the infinite width limit. The main thesis is that the finite-width ReLU network is necessarily superior over the NTK limit, since MKL learning optimizes over all linear combinations over the multiple kernels, but the NTK limit corresponds to a single fixed linear combination of these kernels.

**Strengths:**

The presentation and the contribution of this paper are very straightforward: (convexified gated ReLU) = (MKL). This main insight, in my view, the key and only non-trivial insight of this work. In my view, all other parts of this work are relatively straightforward consequences of this observation, and the experiments are minimal and compact.

Therefore, in my view, the decision comes down to whether this single insight is a sufficient contribution to warrant a NeurIPS publication. In my view, it definitely is. How the hidden convex structure of NN approach compares with the NTK or the mean-field analysis was an important problem to tackle, and I find the insight of this work to be a very good start. (Although this paper doesn't yet deal with the mean-field analysis.) I recommend this paper be accepted.

**Weaknesses:**

.

**Questions:**

.

**Limitations:**

.

---

> ### Author Rebuttal · Authors · 2023-08-09
>
> We would like to thank the reviewer for the positive feedback and accurate comments, and greatly appreciate the time taken by them to review our work. We are encouraged to know that you believe that our main insights warrant an acceptance.

---

### Official Review · Reviewer_98L1 · 2023-07-12

**Soundness:** 2 fair
**Presentation:** 3 good
**Contribution:** 2 fair
**Rating:** 4
**Confidence:** 4

**Summary:**

This paper establishes the equivalence between shallow neural networks activated by ReLU and multiple kernel learning (MKL) through the convex reformulation of shallow neural networks into gated ReLU networks. By interpreting neural networks in this way, it becomes apparent that the network may not be able to attain the optimal MKL kernel on the training set. To address this issue, the authors propose an iterative reweighted method to solve the corresponding MKL problem, which has the potential to obtain the optimal MKL kernel due to its convex nature. Moreover, the authors conduct an analysis of the in-sample prediction error for the gated ReLU networks.

**Strengths:**

Previously, a series of recent studies have demonstrated that (shallow) ReLU networks can be reformulated as a convex problem. This paper builds upon the previous work by establishing the equivalence between this network formulation and the mask kernel learning (MKL) model. This finding has the potential to offer valuable insights into comprehending the performance of neural networks during both training and testing stages.

The paper is well-written, providing an ample and lucid introduction to various topics such as the convex formulation of neural networks, mask kernel learning, NTK, and other related subjects.

**Weaknesses:**

The main contribution of this paper is to establish the equivalence between neural networks and multiple kernel learning (MKL) models. However, there are certain assumptions and setups in the paper that may limit the scope of their findings:

1. The convex formulation of neural networks appears to be applicable only to ReLU (or its variants) activation functions, as the cone constraints rely on the non-negativity of ReLU. Consequently, the results might be specific to ReLU and may not generalize to other activation functions.
2. Since the output layer of the neural network is fully connected, it can be expressed in a feature map format as f(x) = phi(x)^T v, which holds true even for deep neural networks. Thus, merely establishing equivalence to a kernel method may not sufficiently uncover the intricacies of neural networks.
3. In lines 164-167 of the proof for Lemma~3.1, the authors fail to explain why the simplex constraint can be eliminated through a change of variables. This omission could lead to confusion for readers.
4. Theorem 4.3 demonstrates that problem (7) can be reformulated in the form of MKL, but the original problem (7) does not include a simplex constraint. However, the authors do not address this discrepancy in the main theorem or the supplementary proof.
5. The results presented in section 6 do not offer further insights or explain the benefits of formulating the problem as convex optimization or considering it as MKL. This is because the authors solely analyze in-sample prediction, while previous research has shown that shallow ReLU neural networks can generalize well to various underlying distributions (e.g., see https://arxiv.org/abs/1901.08584).

**Questions:**

1. What is the motivation behind using a regularized formulation? Do the main results (Theorem 4.3, Theorem 5.1, Theorem 6.1) hold for the non-regularized version with lambda set to zero?
2. Is it possible to extend the convex formulation to activation functions other than ReLU? If so, what would be the considerations and challenges involved in adapting the formulation to non-ReLU activations?
3. Given that the reformulation is convex, why do you suggest using the Iteratively Reweighted Least Squares (IRLS) algorithm instead of a commonly used convex problem solver? Are there specific benefits to using IRLS in this context? Furthermore, can you provide insights into the properties of the solution that IRLS converges to?

**Limitations:**

See weakness

---

> ### Author Rebuttal · Authors · 2023-08-09
>
> We would like to thank the reviewer for the feedback and comments. We hope that you would consider increasing your score if your concerns are adequately addressed. We have addressed the issues related to generalization in the global response. Please see our responses to your other specific queries below.
>
> $\textbf{Activation functions:}$
> We note that the convex reformulations can be extended to other activation functions. Particularly, for all piecewise linear activations (including ReLU, leaky ReLU, absolute value or binary activations), the same analysis hold except that the structure of the gates (diagonal matrices $\mathbf{D}_i$) changes. We also want to emphasize that the same analysis can also be applied to smooth activations such as sigmoid or tanh but in those cases the computational complexity can be slightly higher due to the increase in the required number of gates/diagonal matrices.
>
> $\textbf{Kernel methods}$
> We are not sure exactly what you mean by the statement that even deep neural networks can expressed in a feature map format. Previous works (apart from the NTK) characterizing deep neural networks establish equivalence to a kernel method by freezing all the weights except for the last layer, effectively resulting in a random features model. The NTK is another kernel characterization, but requires the neural network weights to be in the lazy regime with infinite or large width (where the weights do not change much from initialization). The kernel characterization we present in this work differs from these previous approaches by studying the finite width regime where the weights need not be close to initialization, and the kernel is learnt from the data.
>
> $\textbf{Simplex constraint:}$
> Note that the change of variables argument merely establishes equivalence of problem (12) and the problem described in lines 163-164. We would like to point out that both problems contain the simplex constraint. This constraint, and the weight variable $\eta$ can be eliminated by applying the variational formulation of the squared group $\ell_1$ norm, which is presented in lines 161-162. This allows us to convert the standard $\ell_2$ regularization in the problem between lines 163-164 to the squared group lasso regularization without the weights $\eta$ in problem (14). This is the main result of Lemma 3.1. A further result is the equivalence between $\textit{squared}$ group lasso and the standard group lasso, which is presented in problem (15).
>
> Theorem 4.3 is the result of a chain of equivalences between various optimization problems (proof in the supplement). We present them here again as a summary. (7) is the standard non-convex learning problem for the gated ReLU network. Prior work showed that this is equivalent to the group lasso problem (15). This is also equivalent to the squared group lasso problem (14). Lemma 3.1 then connects (14) to the MKL problem (12). The crux of the argument is as follows: the MKL problem optimizes over parameters $\mathbf{w}$ as well as weights $\eta$ over the kernels, with $\eta$ constrained to be in the simplex, and the parameters $\mathbf{w}$ having standard $\ell_2$ regularization (this is problem (12)). Using the variational argument in the proof of Lemma 3.1, we can completely eliminate the kernel weight variable $\eta$ (along with its simplex constraint), by replacing the standard $\ell_2$ regularization with a squared $\textit{group lasso}$ regularization, which is exactly problem (14).
>
> $\textbf{Regarding Regularization}$
> Yes, the results extend to non-regularized version by taking the limit $\lambda \to 0$ and following the regularization path of the solution. Additionally, we use the regularized objective to avoid issues related to overfitting, leading to better generalization performance in practice.
>
> $\textbf{Why IRLS?}$
> The motivation behind introducing IRLS is that the solution obtained by the NTK can be expressed as one instance of an iterate in the IRLS algorithm, allowing us to directly obtain the optimal MKL solution by initializing with the NTK solution.
>
> Computationally, each iteration of the IRLS method involves a simple least squares regression problem that has a closed form solution which can be computed using very efficient direct and iterative methods. Additionally, state of the art iterative methods like preconditioned conjugate gradient -- specifically LSQR and LSMR [1, 2] that we use for solving these least squares problems can be efficiently parallelized and implemented with GPU acceleration [3].
>
> [1] - Paige, Christopher C., and Michael A. Saunders. "LSQR: An algorithm for sparse linear equations and sparse least squares." ACM Transactions on Mathematical Software (TOMS) 8.1 (1982): 43-71.
>
> [2] - Fong, David Chin-Lung, and Michael Saunders. "LSMR: An iterative algorithm for sparse least-squares problems." SIAM Journal on Scientific Computing 33.5 (2011): 2950-2971.
>
> [3] - Huang, He, et al. "An MPI-CUDA implementation and optimization for parallel sparse equations and least squares (LSQR)." Procedia Computer Science 9 (2012): 76-85.

---

> > ### Comment · Reviewer_98L1 · 2023-08-15
> > **Further Review Comments Based on Author's Rebuttal.**
> >
> > Thank you for the author's response. Having reviewed the rebuttal and comments from other reviewers, I have chosen to maintain my original score due to concerns regarding the applications and implications of this work.
> >
> > I remain unconvinced that the convex reformulation presented in this paper can be readily extended to activations beyond ReLU. As ReLU is also piecewise linear function, I find it lacking that the authors have not provided references or a brief explanation detailing the applicability of such analysis to smoother activations like sigmoid or tanh. Furthermore, neural networks can be considered as a kernel method whenever the output layer is linear. Thus, it is worth considering whether the reformulation and results still hold when the output layer is non-linear, rather than linear.
> >
> > My reservations extend to the empirical results as well. Essentially, [1] reveals that the behavior of neural networks under gradient descent aligns with the kernel gradient with respect to NTK. In the infinite-width scenario, NTK converges to the so-called limiting NTK [1, Theorem 1]. To ensure a fair comparison, numerical experiments should consider juxtapose trained neural networks (across varying widths) with MLK using the optimal MKL kernel obtained in this paper. Unfortunately, based on the provided code, it appears the authors treat MLK with the limiting NTK as the performance of NTK. I hesitate to deem this a fair comparison, as the limiting NTK can be employed in alternative machine learning models like SVM rather than being solely tailored to MLK, but NTK itself really depends on the training.
> >
> > Given my reservations arising from the experimental outcomes, the insights offered by the results in section 6 do not sufficiently underscore the benefits of formulating the problem via convex optimization or viewing it as MKL. Earlier research [2] has indicated that shallow ReLU neural networks exhibit commendable generalization to diverse underlying distributions. Consequently, my decision remains unaltered regarding the review score.
> >
> > [1] Jacot, Arthur, Franck Gabriel, and Clément Hongler. "Neural tangent kernel: Convergence and generalization in neural networks." NeurIPS 2018
> > [2] Arora, Sanjeev, et al. "Fine-grained analysis of optimization and generalization for overparameterized two-layer neural networks." ICML 2019

---

> > > ### Author Response · Authors · 2023-08-18
> > >
> > > We thank you for your detailed response.
> > >
> > > Regarding the activation functions beyond ReLU, we refer the reviewer to [1] for polynomial activation functions (which can sufficiently approximate most other activation functions as well).
> > >
> > > **Regarding Experiments with the NTK**
> > >
> > > We are not exactly sure by what you mean when you say that the experiments are not a fair comparison. Specifically, our experiments do indeed include comparisons with finite width neural networks trained by gradient descent. In Figure 1, the learned functions with the black dashed lines are the result of training a finite width neural network via gradient descent. We observe that the convex MKL formulation better matches this result than the limiting NTK. Could you please clarify what exactly you mean when you refer to "MLK with the limiting NTK as the performance of the NTK"? Additionally, the MKL approach can also be extended to SVMs and other applications where standard kernel methods can be applied [2].
> > >
> > > [1] - Bartan, Burak, and Mert Pilanci. "Neural spectrahedra and semidefinite lifts: Global convex optimization of polynomial activation neural networks in fully polynomial-time." arXiv preprint arXiv:2101.02429 (2021).
> > >
> > > [2] - Bach, Francis R., Gert RG Lanckriet, and Michael I. Jordan. "Multiple kernel learning, conic duality, and the SMO algorithm." Proceedings of the twenty-first international conference on Machine learning. 2004.

---

> ### Author Response · Authors · 2023-08-20
>
> Dear Reviewer 98L1,
>
> We believe that we have addressed your concerns in our responses. Since the deadline is approaching, we would like to hear your feedback so that we can respond to that before the discussion period ends. Please feel free to raise questions if you have other concerns. Thank you very much for your support, we really appreciate that!
>
> Best regards,
>
> Authors

---

> > ### Comment · Reviewer_98L1 · 2023-08-20
> >
> > After reviewing the code in the 'UCI' directory, it is evident that Table 1 is generated using this codebase. However, it's important to note that the reported test accuracy for the NTK does not originate from a conventionally trained neural network. Instead, it seems to be derived from MLK with the so-called limiting NTK. It is reasonable to expect that MLK with a restricted limiting NTK might exhibit inferior performance compared to MLK with the optimal kernel. That is why I feel the experiments are not so fair.
> >
> > To obtain a more comprehensive assessment of test (and train) performance, I recommend the authors consider conducting experiments involving neural networks with varying widths which is used in [1]. By comparing the accuracies across different network configurations, we can draw more meaningful conclusions. Specifically, if most of these test accuracies fall below the performance achieved with MLK using the optimal kernel, it would support the assertion that MLK outperforms NTK in terms of test performance.
> >
> > [1] Lee, J., Bahri, Y., Novak, R., Schoenholz, S. S., Pennington, J., & Sohl-Dickstein, J. (2018, February). Deep Neural Networks as Gaussian Processes. In International Conference on Learning Representations.

---

> > > ### Author Response · Authors · 2023-08-21
> > >
> > > We thank the reviewer for additional comments. We first would like to emphasize that our main objective in this work is to study the optimization of regularized neural networks through a novel kernel perspective. Thus, we don't provide any theory or an extensive set of experiments regarding the generalization properties of the proposed approach which itself requires a comprehensive analysis that can be the focus of a single paper.
> > >
> > > Additionally, we would like to clarify that the NTK is itself a kernel method which aims to approximate the conventional neural network training procedure, using a kernel derived from the infinite width limit of the neural network. This is in contrast to our novel kernel characterization, the convex MKL approach, which is also a kernel method, but with an optimal kernel that is learnt from the data. Note that we do not require the infinite width or large width assumption for this kernel characterization to hold. Our theory shows that this MKL approach is equivalent to the conventional neural network on the training set. In addition to this theory, we present empirical results (in Table 1 and Fig 1) which compare the test performances of the two kernel characterizations, and show that our convex MKL formulation achieves better test performance than the NTK formulation on multiple datasets. Since we are empirically comparing two different kernel methods, we believe that this is a fair comparison. Specifically, we perform standard kernel ridge regression (KRR) with the NTK kernel to obtain the test accuracies for the NTK in Table 1. Note that we do not perform "MLK" for this column (nor is it clear what this exactly means). Similarly, to obtain the test accuracies for our novel kernel characterization in Table 1, we solve the MKL problem with masking kernels that we derived from the convex reformulation of the gated ReLU network.
> > >
> > > We are not entirely sure how running conventional neural network training would show that the NTK is suboptimal, since the NTK is not an accurate approximation of conventional neural network training when the width is not large or close to infinity. Could you please clarify the details of the experimental setup you are proposing that would lead to a more conclusive comparison between the two kernel characterizations?

---

> > > > ### Comment · Reviewer_98L1 · 2023-08-21
> > > >
> > > > Dear Authors,
> > > >
> > > > I wanted to provide some clarity on the concepts we've been discussing. As described in [1], the Neural Tangent Kernel (NTK) serves as a kernel that characterizes how neural networks evolve under gradient descent with an infinitesimal learning rate (gradient flow). The formal definition of NTK $\Theta^{(L)}(\theta)$ is introduced at the beginning of Section 4 [1]. Notably, this NTK $\Theta^{(L)}(\theta)$ is defined for any neural network trained under gradient descent, regardless of its width; it's not limited to networks with infinite width. Furthermore, $\Theta^{(L)}(\theta)$ evolves continuously during training as parameters $\theta$ are updated.
> > > >
> > > > In contrast, as outlined in [Theorem1, 1], under the assumption of $n_{L}=1$ in the infinite-width limit, NTK $\Theta^{(L)}$ converges in probability to a fixed deterministic kernel $\Theta_{\infty}^{(L)}$. This deterministic limiting kernel is commonly known as the **limiting NTK** in the literature. It was the limiting NTK used in your experiments instead of the NTK. In summary, NTK $\Theta^{(L)}(\theta)$ applies to all neural networks trained via gradient descent (regardless of width) and changes continuously during training, while the limiting NTK $\Theta^{(L)}_{\infty}$ is a fixed kernel in the infinite-width limit.
> > > >
> > > > For the new experiment design, I recommend consulting the framework introduced in [2]. Similarly, [2] establishes a connection between neural networks and another kernel method known as Gaussian processes. Specifically, as the width of a neural network approaches infinity, it converges toward behaving like a Gaussian process, termed the "Neural Network and Gaussian Process (NNGP) correspondence." This correspondence is associated with a unique NNGP kernel, distinct from the NTK kernel. To explore their theoretical findings, they compare the **trained** neural networks with various widths to GPs equipped with the NNGP kernel. Figure 1 in their work illustrates the comparison in terms of test accuracy, and it could be straightforward to extend this comparison to training accuracy.
> > > >
> > > > I appreciate your patience and hope these explanations provide a clearer picture of the concepts we've been discussing.
> > > >
> > > > [1] Jacot, Arthur, Franck Gabriel, and Clément Hongler. "Neural tangent kernel: Convergence and generalization in neural networks."  NeurIPS 2018
> > > >
> > > > [2] Lee, J., Bahri, Y., Novak, R., Schoenholz, S. S., Pennington, J., & Sohl-Dickstein, J. (2018, February). Deep Neural Networks as Gaussian Processes. ICLR 2018

---

### Author Rebuttal · Authors · 2023-08-09

We would like to thank all the reviewers and the AC for taking the time to review and assess our work.

We are encouraged to know that the reviewers believe that our findings are valuable and our main insights are sufficiently novel. We also appreciate that the reviewers found the paper well written and lucid, as clarity of presentation is important to us.

In this global response, we address some of the common queries and concerns brought up by the reviewers.

$\textbf{Regarding Generalization:}$

We would like to emphasize that the focus of this paper is on optimization of neural networks with a standard regularized training objective. Indeed, theoretically studying the generalization properties of the MKL kernel and convex reformulations is a very interesting direction for future work, but the focus of our theoretical results in this work is on the training performance. **With regards to generalization, we present empirical results that indicate that our convex MKL formulation also achieves better test performance than the NTK formulation on multiple datasets (Table 1 for UCI datasets, and the learned functions in Figure 1 for toy datasets).**


$\textbf{Complexity arising from number of hyperplane arrangements/kernels:}$

There are multiple ways to avoid high computational complexity as detailed below.

* First, one can use a sampling based approach where one can randomly sample a tiny subset of all possible hyperplane arrangements and then solve the convex program with this subset. Thus, although the resulting approach isn't exact, **the training complexity won't be exponential in  any of the problem parameters anymore**. The experimental results in Section 7 show that this approximation in fact works extremely well, specifically resulting in models that outperform the NTK in 26/33 UCI datasets.

* Second, we can change the architecture. Particularly, we can replace fully connected networks with convolutional networks. Then, since CNNs operate on the patch matrices $\\{\mathbf{X}_b\\}\_{b=1}^B$ instead of the full data matrix $\mathbf{X}$, where $\mathbf{X}\_b \in \mathbb{R}^{n \times h}$ and $h$ denotes the filter size, even when the data matrix is full rank, i.e., $r=\min (n,d)$, the number of hyperplane arrangements $p$ is upperbounded as $p \leq \mathcal{O}(n^{ r_c})$, where $r_c:=\max_b \mathrm{rank}(\mathbf{X}_b)\leq h \ll \min(n,d)$. For instance, let us consider a CNN with $3 \times 3$ filters, then $r_c \leq 9$ independent of $n,d$. As a consequence, weight sharing structure in CNNs dramatically limits the number of possible hyperplane arrangements and avoids exponential complexity. This also supports the observed efficiency and remarkable generalization performance of CNNs in practice.

---

### Decision · Program_Chairs · 2023-09-21

**Decision:**

Accept (poster)

**Comment:**

The theoretical results in this paper are correct and interesting, which reveal whether DNN training can be interpreted as kernel learning. The authors need to polish and improve the paper according to the reviewers' suggestions in the revision.